# An End-to-End Model for Logits-Based Large Language Models Watermarking

**Kahim Wong** [1]   **Jicheng Zhou** [1]   **Jiantao Zhou** [1]   **Yain-Whar Si** [2]

## Abstract

The rise of LLMs has increased concerns over source tracing and copyright protection for AIGC, highlighting the need for advanced detection technologies. Passive detection methods usually face high false positives, while active watermarking techniques using logits or sampling manipulation offer more effective protection. Existing LLM watermarking methods, though effective on unaltered content, suffer significant performance drops when the text is modified and could introduce biases that degrade LLM performance in downstream tasks. These methods fail to achieve an optimal tradeoff between text quality and robustness, particularly due to the lack of end-to-end optimization of the encoder and decoder. In this paper, we introduce a novel end-to-end logits perturbation method for watermarking LLM-generated text. By joint optimization, our approach achieves a better balance between quality and robustness. To address non-differentiable operations in the end-to-end training pipeline, we introduce an online-prompting technique that leverages the on-the-fly LLM as a differentiable surrogate. Our method achieves superior robustness, out-performing distortion-free methods by 37–39% under paraphrasing and 17.2% on average, while maintaining text quality on par with the distortion-free methods in terms of text perplexity and downstream tasks. Our method can be easily generalized to different LLMs. Code is available at https://github.com/KAHIMWONG/E2E_LLM_WM.

[1]State Key Laboratory of Internet of Things for Smart City, Department of Computer and Information Science, Faculty of Science and Technology, University of Macau, China [2]Department of Computer and Information Science, Faculty of Science and Technology, University of Macau, China. Correspondence to: Jiantao Zhou <jtzhou@um.edu.mo>.

*Proceedings of the 42nd International Conference on Machine Learning*, Vancouver, Canada. PMLR 267, 2025.

## 1. Introduction

Large Language Models (LLMs) like ChatGPT (Achiam et al., 2023), Llama (Touvron et al., 2023; Dubey et al., 2024), and OPT (Zhang et al., 2022) have greatly improved the quality of AI-generated content (AIGC), broadening their applications in fields such as translation (Hendy et al., 2023), content creation (Ni et al., 2023), etc. However, such a rapid expansion has also raised concerns about copyright infringement, academic dishonesty, and unethical practices (Augenstein et al., 2024). These issues highlight the urgent need for reliable methods to distinguish between human-written text and LLM-generated content, ensuring digital integrity and combating misinformation (Barrett et al., 2023).

Numerous approaches have been proposed to address LLM ethical concerns by detecting LLM-generated content. Passive detection methods focus on identifying unique properties of generated text, often through training binary classifiers (Bakhtin et al., 2019; Jawahar et al., 2020) or statistical techniques like DetectGPT (Mitchell et al., 2023), which compares a sentence's log-probability to that of a perturbed version. However, as LLMs improve and the gap between generated and human-written text narrows, the effectiveness of these methods declines dramatically. In contrast, active detection methods, like embedding watermarks in generated text, are proving to be more robust alternatives. LLM watermarking methods fall into two main categories: logits-based and sampling-based. Specifically, logits-based methods (Kirchenbauer et al., 2023; Liu et al., 2024b; Huo et al., 2024) randomly divide the vocabulary into "green" and "red" lists by hashing preceding tokens as the seed. Then, perturbations are introduced to the logits that favor green list tokens in the generated text, and the proportion of the "green" tokens is used to distinguish whether a text is LLM-generated. Sampling-based methods (Kuditipudi et al., 2024; Christ et al., 2024) rely on random bitstream to guide token sampling, creating detectable correlations in the text. Despite advancements, current watermarking schemes experience significant performance degradation with even slight text modifications. Additionally, existing algorithms introduce logit biases or guide sampling through random bitstream, which would result in semantic differences between watermarked and non-watermarked content,

negatively impacting LLM performance on downstream tasks, and limiting the practicality of these watermarking techniques. Among logits-based methods, there is a growing trend to replace the hashing schemes with trainable networks for generating logit perturbations (Liu et al., 2024b; Huo et al., 2024), and to substitute statistical decoding with trainable decoders (Liu et al., 2024a), leveraging the flexibility of neural network training to improve performance. However, these existing approaches still fail to achieve an optimal trade-off between text quality and robustness, primarily due to the separate training of the encoder and decoder rather than a joint, end-to-end optimization.

With this observation, we introduce a novel logits-based end-to-end model, where lightweight encoder and decoder networks are jointly optimized to enhance both detecting robustness and text quality. Note that building such an end-to-end system is highly non-trivial because many involved modules, such as the complex text modification and the semantic loss computation, are inherently non-differentiable. To resolve these challenges brought by the non-differentiability, we introduce a novel online prompting technique that utilizes the on-the-fly LLM as a differentiable surrogate. This approach enables the model to effectively handle the above-mentioned non-differentiable operations. Our framework is LLM-agnostic, allowing any LLM to be used during training, and once trained, the model can be applied to other LLMs without retraining.

Our key contributions are as follows:

- We present a novel logits-based end-to-end model for LLM watermarking, improving detection robustness and text quality through encoder-decoder joint optimization.

- We introduce a new online prompting technique that transforms non-differentiable operations, such as semantic loss calculation and advanced online text modification, into differentiable operations by dynamically prompting the on-the-fly LLM. This technique enables seamless end-to-end training without external models.

- Extensive experiments show that our method achieves superior robustness, outperforming distortion-free methods by 37–39% under paraphrasing and 17.2% on average, while maintaining text quality on par with these distortion-free methods in terms of PPL and downstream tasks. Notably, our method can be generalized across LLMs without additional training.

The paper is organized as follows: Sec. 2 reviews related works on LLM watermarking. Sec. 3 details our proposed method. Sec. 4 presents experimental results demonstrating the superior performance of our method. Finally, Sec. 5 concludes the paper.

## 2. Related Works on LLM Watermarking

Recent works on LLM watermarking can be classified into two main categories: logits-based and sampling-based methods. Logits-based methods, as presented in KGW (Kirchenbauer et al., 2023), split the vocabulary into "green" and "red" lists by hashing preceding tokens and biasing green list logits to favor their selection, using the proportion of green tokens to detect watermarks. Building on KGW, several methods aim to improve the robustness, quality, or unforgeability of the watermarked text. KGW-R (Kirchenbauer et al., 2024) explores different hashing schemes, while Unigram (Zhao et al., 2024) uses a fixed red-green separation to enhance editing resistance. SWEET (Lee et al., 2023) selectively modifies logits at high-entropy tokens to improve quality in low-entropy scenarios such as the code generation. UPV (Liu et al., 2024a) employs an encoder network to split lists and a detector network for classification, enabling public detection. SIR (Liu et al., 2024b) trains an encoder to apply context-aware biases for better robustness, and TSW (Huo et al., 2024) uses two networks to adaptively adjust watermark strength and list-splitting ratio for a better balance between detectability and text quality. DiPmark (Wu et al., 2024) enhances selected token probabilities by applying a distribution-preserving reweight function. Sampling-based methods, such as EXP (Kuditipudi et al., 2024), use a pseudo-random bitstream to guide token selection through inverse sampling. This process produces watermarked text that aligns with the bit sequence and makes detection accurate. EXP-Edit (Christ et al., 2024) builds on the EXP method by introducing edit distance to measure sequence alignment, enhancing robustness against tampering. Unbiased (Zhao et al., 2024) employs inverse sampling and permutation-based reweighting for watermarking but relies on LLM API logits and generation prompts, reducing its efficiency. Nevertheless, as will be shown, current LLM watermarking methods suffer from significant drops in detection performance when the text is modified and can introduce biases that impair LLM performance, such as the translation and the code generation.

## 3. Method

In this section, we detail our LLM watermarking solution, which improves the balance between robustness and text quality, compared to existing methods. We begin by outlining LLM and logits-based watermarking basics, followed by the structure and objectives of our model and the online prompting technique. Finally, we introduce a plug-and-play converter for cross-LLM inference.

First, let us give a brief overview of the LLM workflow and the logits-based method. Given a prompt $\mathbf{X}_{prompt} = [\mathbf{x}_1, \ldots, \mathbf{x}_k]$, an LLM $M$ generates tokens in an autoregressive manner. At each time step $t$, $M$ produces a proba-

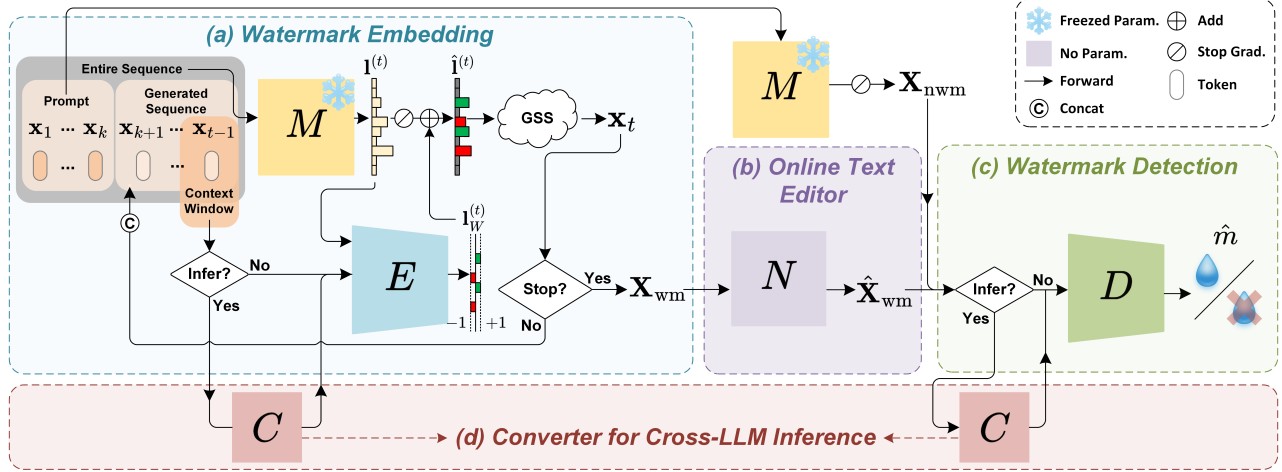

*Figure 1.* Overview of our end-to-end model, consisting of (a) watermarking via logits perturbation with encoder $E$; (b) simulating user edits using the online text editor $N$; and (c) detecting watermarked content through decoder $D$. The entire model is trained end-to-end to optimize both the quality and detection accuracy of the watermarked content. In the inference phase, (d) a converter is deployed for cross-LLM adaption. GSS is the abbreviation of the Gumbel-Softmax sampling.

bility distribution for the next token $p_M(\mathbf{x}_t|\mathbf{x}_1,\ldots,\mathbf{x}_{t-1})$ over vocabulary $\mathcal{V} = \{\mathbf{t}_1,\ldots,\mathbf{t}_{|\mathcal{V}|}\}$, then the token $\mathbf{x}_t$ is sampled from $p_M(\mathbf{x}_t)$ . In this process, logits $\mathbf{l}^{(t)} = [l_1^{(t)},\ldots,l_{|\mathcal{V}|}^{(t)}] \in \mathbb{R}^{|\mathcal{V}|}$ refer to the unnormalized output by $M$ before converting into probability

$$p_M(\mathbf{x}_t = \mathbf{t}_k|\mathbf{x}_1,\ldots,\mathbf{x}_{t-1}) = \frac{\exp(l_k^{(t)})}{\sum_{i=1}^{|\mathcal{V}|}\exp(l_i^{(t)})}. \quad (1)$$

Once the stop criteria are satisfied, the user receives a generated response $\mathbf{X}_{\text{nwm}}$. To embed a watermark into the generated text, logits-based methods introduce a watermark logits $\mathbf{l}_W^{(t)}$ at each generation step with a strength $\delta$, resulting in the perturbed logits: $\hat{\mathbf{l}}^{(t)} = \mathbf{l}^{(t)} + \delta \cdot \mathbf{l}_W^{(t)}$. By adjusting the logits to favor tokens in the green list, which is determined by hashing preceding tokens, the method increases the probabilities of sampling green-list tokens. Consequently, the watermarked text $\mathbf{X}_{\text{wm}}$ contains a higher proportion of these favored tokens, creating a detectable statistical cue that is not present in human-written content.

We are now ready to present the details of our end-to-end watermark method.

### 3.1. Model Overview

The architecture of our proposed method is depicted in Fig. 1. At each time step $t$, the encoder $E$ takes the context $\mathbf{C}^{(t)} = [\mathbf{x}_{t-1-w},\ldots,\mathbf{x}_{t-1}]$ from a local window $W$ and the current logits $\mathbf{l}^{(t)}$ from the on-the-fly LLM $M$ to generate the watermark logits $\mathbf{l}_W^{(t)} = E(\mathbf{C}^{(t)},\mathbf{l}^{(t)})$. The next token is then generated with the perturbed logits $\hat{\mathbf{l}}^{(t)} = \mathbf{l}^{(t)} + \delta \cdot \mathbf{l}_W^{(t)}$ based on Gumbel-Softmax sampling (GSS) to enable differentiable sampling for end-to-

end training. Once the stop criteria are reached, the entire watermarked sequence $\mathbf{X}_{\text{wm}}$ is generated, and the online text editor $N$ augments the text to simulate user edits, resulting in $\hat{\mathbf{X}}_{\text{wm}} = N(\mathbf{X}_{\text{wm}})$, which is then passed to the decoder $D$ to detect the presence of the watermark $\hat{\mathbf{m}} = D(\hat{\mathbf{X}}_{\text{wm}}) \in \{0,1\}$, where "0" indicates no watermark and "1" indicates the presence of the watermark. The same prompt $\mathbf{X}_{\text{prompt}}$ is fed into the standard LLM pipeline with $M$ to generate the non-watermarked sample $\mathbf{X}_{\text{nwm}}$ for $D$. The networks are jointly trained, with $E$ updated via backpropagation from $D$. After the model is trained, a converter $C$ is appended before both $E$ and $D$ for cross-LLM inference.

### 3.2. Watermark Embedding

We illustrate the watermark embedding process of our method, as shown in Fig. 1 (a) colored with light blue. Similar to KGW, our approach embeds a watermark on the generated text by biasing the original logits. Instead of using context token hashing to determine the bias, we employ a lightweight network $E$ to implicitly learn watermark logits by minimizing the detection loss $\mathcal{L}_{\text{dec}}$ for $D$ and the semantic loss $\mathcal{L}_{\text{sem}}$ between $\mathbf{X}_{\text{wm}}$ and $\mathbf{X}_{\text{nwm}}$. However, constructing such an end-to-end pipeline poses challenges due to the non-differentiable modules involved, including the token sampling process, the online editing of $\mathbf{X}_{\text{wm}}$, and the computation of $\mathcal{L}_{\text{sem}}$. To address this, we implement several alternatives, including using GSS to replace the non-differentiable sampling. Additionally, we propose an online prompting technique to perform online editing of $\mathbf{X}_{\text{wm}}$, as well as extract semantic embeddings to compute $\mathcal{L}_{\text{sem}}$ by dynamically prompting the on-the-fly LLM.

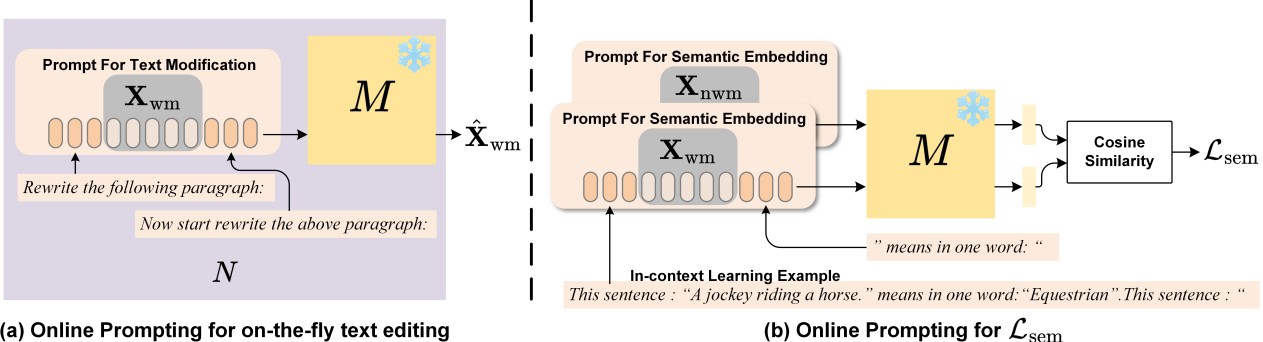

(a) Online Prompting for on-the-fly text editing

(b) Online Prompting for $\mathcal{L}_{\text{sem}}$

*Figure 2.* Online prompting technique for (a) computing semantic loss and (b) on-the-fly text editing. The prompts are first converted into the embeddings and then concatenated with the generated text $\mathbf{X}_{\text{wm}}$ and $\mathbf{X}_{\text{nwm}}$.

We now formulate the data flow of the watermark embedding process. At each time step $t$, the encoder $E$ receives the context $\mathbf{C}^{(t)}$ and the current logits $\mathbf{l}^{(t)}$ to generate the watermark logits $\mathbf{l}_W^{(t)}$ across the vocabulary $\mathcal{V}$. The encoder considers all possible token sequences $\mathbf{S}^{(t)} = [\mathbf{S}_1^{(t)}, \ldots, \mathbf{S}_{|\mathcal{V}|}^{(t)}]$, where each sequence is formed by $\mathbf{S}_i^{(t)} = [\mathbf{C}^{(t)}, \mathbf{t}_i]$. Nevertheless, due to the large size of $\mathcal{V}$, processing all sequences is computationally expensive. For efficiency, we focus on the top-$k$ logits tokens $\mathbf{t}_{i_1}, \ldots, \mathbf{t}_{i_k}$, and form the top-$k$ sequences $\mathbf{S}_{\text{top-}k}^{(t)} = [\mathbf{S}_{i_1}^{(t)}, \ldots, \mathbf{S}_{i_k}^{(t)}]$, in which $\{i_1, i_2, \ldots, i_k\}$ is the index of the top-$k$ logits tokens. A multilayer perceptron (MLP) $f_{\text{mlp}}$ maps the input sequence $\mathbf{S}_{\text{top-}k}^{(t)}$ to the watermark logits of the top-$k$ tokens:

$$\mathbf{l}_{\text{top-}k}^{(t)} = \tanh(\tau_t \cdot f_{\text{mlp}}(\mathbf{S}_{\text{top-}k}^{(t)})), \qquad (2)$$

where $\tanh$ bounds the output within $[-1, 1]$ and the parameter $\tau_t$ adjusts the sharpness of $\tanh$. As visualized by $\mathbf{l}_W^{(t)}$ in Fig. 1 (a), with logits of token close to -1 belonging to the "red" list and 1 indicating the "green" list. Except for the top-$k$ tokens, the watermark logits of the remaining tokens (can be considered in the "grey" list) are padded with 0 to form $\mathbf{l}_W^{(t)}$.

By adding up watermark logits on the original logits with strength $\delta$, we obtain the perturbed logits $\hat{\mathbf{l}}^{(t)}$. Then, GSS (Jang et al., 2017) is used to allow the sampling step differentiable, mimicking the standard LLM sampling and enabling end-to-end training. Specifically, Gumbel noise $g_i = -\log(-\log(U_i))$, where $U_i \sim \text{Uniform}(0, 1)$, is added to each logit, yielding $\tilde{l}_i^{(t)} = \hat{l}_i^{(t)} + g_i$. The logits $\tilde{\mathbf{l}}^{(t)}$ are then passed through the softmax function:

$$\mathbf{p}_M(\mathbf{x}_t) = \frac{\exp((\hat{\mathbf{l}}^{(t)} + \mathbf{g})/\tau_g)}{\sum_{i=1}^{V} \exp((\hat{l}_i^{(t)} + g_i)/\tau_g)}, \qquad (3)$$

where $\mathbf{g}$ is a vector of Gumbel noise with the same shape as $\hat{\mathbf{l}}^{(t)}$ and $\tau_g$ controls the sharpness of the softmax. The next token embedding is computed as $\mathbf{x}_t = \mathbf{p}_M^T \mathbf{E}$, where $\mathbf{E}$ is

the token embedding matrix of $\mathcal{V}$. The watermark signal is embedded at each generation step until the stop criteria are met and the watermarked sample $\mathbf{X}_{\text{wm}}$ is obtained.

### 3.3. Online Text Editing

As shown in Fig. 1 (b), the online text editor $N$ (in light purple) is positioned between the encoder and decoder during training to simulate user edits of watermarked content, enhancing detection robustness. We utilize the online prompting technique in Fig. 2 (a) to augment watermarked text by dynamically prompting the online LLM $M$. This approach effectively handles non-differentiable online text modifications, such as rewriting. Specifically, after generating watermarked text $X_{wm}$, it is fed into N to produce $\hat{X}_{wm}$. The editing prompt: *Rewrite the following paragraph:* [text] *. Now start to rewrite the above paragraph:* instructs $M$ to generate an augmented version of the watermarked content, where [text] is a placeholder for $\mathbf{X}_{\text{wm}}$. Thus, $N(\mathbf{X}_{\text{wm}})$ is equivalent to $M([\mathbf{X}_{\text{epb}}, \mathbf{X}_{\text{wm}}, \mathbf{X}_{\text{epe}}])$ in which $\mathbf{X}_{\text{epb}}$ and $\mathbf{X}_{\text{epe}}$ denote the beginning and the end of the editing prompt. In contrast to existing online text editing methods, such as random token dropping/adding (Zhang et al., 2024), our approach shows significantly improved robustness to unseen distortions due to the capabilities of the on-the-fly $M$ to perform more complex modification on the text (more examples are provided in Appendix C.4). While external paraphrasing models like Dipper (Krishna et al., 2023) are available, differences in tokenizers between the LLM and these models render the operation non-differentiable.

### 3.4. Watermark Detection

As shown in Fig. 1 (c), a lightweight network $D$ (in green) is used to classify whether a given text is watermarked or not, allowing end-to-end training for more robust detection compared to purely statistical methods. $\hat{\mathbf{X}}_{\text{wm}}$ (altered or unaltered) and $\mathbf{X}_{\text{nwm}}$ are fed into $D$, which predicts whether the text contains a watermark $\hat{m} = D(\mathbf{X})$. For efficiency, $D$

is built with LSTM layers and an MLP classification head.

### 3.5. Training Objectives

The entire end-to-end system is supervised with two objectives: detection loss $\mathcal{L}_{\text{dec}}$ and semantic loss $\mathcal{L}_{\text{sem}}$. The detection loss $\mathcal{L}_{\text{dec}}$, which is computed using cross-entropy between the prediction and the ground-truth label to detect the watermark signal accurately. Additionally, to preserve the capabilities of the original LLM, $\mathcal{L}_{\text{sem}}$ ensures that $\mathbf{X}_{\text{wm}}$ retains the same semantics as $\mathbf{X}_{\text{nwm}}$. Semantic loss can be typically computed by the distance between embeddings which are extracted from an external semantic model $f_{\text{sem}}$, such that $\mathbf{e}_{\text{nwm}} = f_{\text{sem}}(\mathbf{X}_{\text{nwm}})$ and $\mathbf{e}_{\text{wm}} = f_{\text{sem}}(\mathbf{X}_{\text{wm}})$, and the loss can be computed by

$$\mathcal{L}_{\text{sem}}(\mathbf{e}_{\text{nwm}}, \mathbf{e}_{\text{wm}}) = 1 - \frac{\langle \mathbf{e}_{\text{nwm}}, \mathbf{e}_{\text{wm}} \rangle}{\|\mathbf{e}_{\text{nwm}}\|_2 \|\mathbf{e}_{\text{wm}}\|_2}. \quad (4)$$

Still, extracting the embeddings $\mathbf{e}_{\text{nwm}}$ and $\mathbf{e}_{\text{wm}}$ is non-differentiable. Unless the online LLM $M$ and the semantic model $f_{\text{sem}}$ share the same tokenizer, the embedding mapping between the two is not bijective, complicating the transformation from $M$ to the $f_{\text{sem}}$ domain. To address the issue, we avoid external models for computing semantics and instead prompt the on-the-fly LLM to approximate $f_{\text{sem}}$. Fortunately, recent studies have shown that LLMs can generate semantic embeddings with quality comparable to dedicated semantic models using prompt engineering, without the need for fine-tuning (Jiang et al., 2023). We leverage this LLM ability to compute $\mathcal{L}_{\text{sem}}$ directly, enabling end-to-end training and resolving non-differentiability issues. As illustrated in Fig. 2 (b), $\mathbf{X}_{\text{wm}}$ and $\mathbf{X}_{\text{nwm}}$ are generated in parallel at the generation phase, and a prompt with an in-context learning example: *This sentence: "A jockey riding a horse." means in one word: "Equestrian". This sentence: "*[text]*" means in one word: "* guides the on-the-fly LLM to extract semantic embeddings. [text] is the placeholder for $\mathbf{X}_{\text{nwm}}$ or $\mathbf{X}_{\text{wm}}$, and the embeddings $\mathbf{e}_{\text{nwm}} = M([\mathbf{X}_{\text{spb}}, \mathbf{X}_{\text{nwm}}, \mathbf{X}_{\text{spe}}])$ and $\mathbf{e}_{\text{wm}} = M([\mathbf{X}_{\text{spb}}, \mathbf{X}_{\text{wm}}, \mathbf{X}_{\text{spe}}])$ are extracted, where $\mathbf{X}_{\text{spb}}$ and $\mathbf{X}_{\text{spe}}$ are the beginning and the end of the above prompt. Finally, $\mathcal{L}_{\text{sem}}$ is computed by the cosine distance between the two embeddings as formulated in Eq. (4).

### 3.6. Cross-LLM Inference

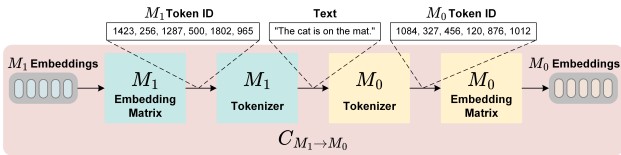

*Figure 3.* Converter for cross-LLM Inference.

After training $E^*_{M_0}$ and $D^*_{M_0}$ on a specific LLM $M_0$ using the end-to-end pipeline, we consider the generalizability of our method to other LLMs. Due to unique tokenizers and embedding dimensions, $E^*_{M_0}$ and $D^*_{M_0}$ cannot be directly applied to another LLM $M_1$. To address this, we introduce a converter $C$ that transforms embeddings from $M_1$ to $M_0$, as located in Fig. 1 (d) and detailed in Fig. 3. This process involves converting $M_1$ embeddings to the text domain and then back to $M_0$ embedding space. The converter is positioned before $E^*_{M_0}$ and $D^*_{M_0}$, enabling cross-LLM watermark embedding and detection as $E^*_{M_0}(C_{M_1 \to M_0}(\mathbf{S}))$ and $D^*_{M_0}(C_{M_1 \to M_0}(\mathbf{X}))$, respectively. Our encoder accepts a fixed number of context tokens, requiring careful token management due to the variability in token segmentation across different tokenizers. To ensure that the transformed context tokens exceed the original length, we dilate the context window $W$ by a factor of 2 and then truncate the latest context to maintain the appropriate length $W$.

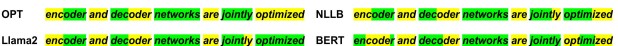

*Figure 4.* Visualize the tokenization of *OPT-1.3B* (Zhang et al., 2022), *Llama2-7B* (Touvron et al., 2023), *NLLB-600M* (Costa-jussà et al., 2022), and *BERT-base* (Kenton & Toutanova, 2019) tokenizers. Each single token is indicated with a color block alternatively.

To analyze the applicability of converter $C$ for cross-LLM inference, we visualize the tokenization of 4 tokenizers in Fig. 4, and observe that the tokenizers often decompose sentences similarly. The converter $C$ processes candidate sequences $\mathbf{S}^{(t)}$ (for the encoder) and sentence $\mathbf{X}$ (for the decoder), which $\mathbf{S}^{(t)}$ include the context $\mathbf{C}^{(t)}$ and the next token candidate $\mathbf{t}_i$. Analysis of the transformation reveals that $\mathbf{X}$ and $\mathbf{C}^{(t)}$, containing numbers of tokens, can tolerate token variations. The transformed $\mathbf{t}_i$ can result in three outcomes: 1) merging with the preceding tokens (e.g., "der" merges with "deco" to form "decoder"); 2) splitting into sub-tokens (e.g., "jointly" becomes "joint" and "ly"); and 3) remaining unchanged. While the third outcome is ideal, cases 1 and 2 may negatively impact performance of the watermark model.

*Table 1.* Normalized Levenstein similarity between tokenized sentences across 4 tokenizers.

| Tokenizer | OPT-1.3B | Llama2-7B | NLLB-600M | BERT-base |
|---|---|---|---|---|
| *OPT-1.3B* | 1.000 | 0.711 | 0.723 | 0.838 |
| *Llama2-7B* | - | 1.000 | 0.730 | 0.681 |
| *NLLB-600M* | - | - | 1.000 | 0.721 |

To estimate the occurrence probability of cases 1 and 2, we quantify the alignment using normalized Levenshtein similarity between token lists for the same sentences across tokenizers, as shown in Table 1. The average token align-

*Table 2.* Performance of the LLM watermark methods. CL: Clean watermark sample; SS: Synonymous substitution; CP: Copy-paste attack; PA: Paragraphing; PPL: text perplexity; $\Delta_{\text{Unigram}}$: Unigram vs. Ours; $\Delta_{\text{DiPmark}}$ DiPmark vs. Ours; NWM: No watermark.

| Method | OPT-1.3B | | | | | Llama2-7B | | | | | Qwen2.5-7B | | | | |
| | Robustness (F1↑) | | | | Quality | Robustness (F1↑) | | | | Quality | Robustness (F1↑) | | | | Quality |
| | CL | SS | CP | PA | PPL↓ | CL | SS | CP | PA | PPL↓ | CL | SS | CP | PA | PPL↓ |
|---|---|---|---|---|---|---|---|---|---|---|---|---|---|---|---|
| NWM | - | - | - | - | 10.484 | - | - | - | - | 6.811 | - | - | - | - | 8.921 |
| KGW | 1.000 | 0.990 | 0.983 | 0.880 | 13.173 | 1.000 | 0.970 | 0.846 | 0.858 | 8.658 | 1.000 | 0.983 | 0.975 | 0.832 | 11.419 |
| Unigram | 1.000 | 0.997 | 0.943 | 0.943 | 12.739 | 0.995 | 0.990 | 0.873 | 0.909 | 9.275 | 1.000 | 0.993 | 0.955 | 0.942 | 10.847 |
| Unbiased | 0.992 | 0.800 | 0.949 | 0.680 | 11.940 | 0.990 | 0.785 | 0.912 | 0.684 | 7.565 | 0.985 | 0.780 | 0.930 | 0.683 | 10.061 |
| DiPmark | 0.997 | 0.809 | 0.954 | 0.692 | 12.085 | 0.983 | 0.779 | 0.915 | 0.670 | 7.681 | 0.985 | 0.780 | 0.923 | 0.681 | 10.488 |
| Ours | 0.998 | 0.992 | 0.975 | 0.952 | 12.397 | 0.995 | 0.985 | 0.978 | 0.916 | 7.730 | 0.995 | 0.985 | 0.985 | 0.945 | 9.997 |
| $\Delta_{\text{Unigram}}$ | 0% | -1% | +3% | +1% | +3% | 0% | 0% | +12% | +1% | +17% | -1% | -1% | +3% | 0% | +8% |
| $\Delta_{\text{DiPmark}}$ | 0% | +23% | +2% | +38% | -3% | +1% | +26% | +7% | +37% | -1% | +1% | +26% | +7% | +39% | +5% |

ment probability is 73.4%. We find that given a large enough number of candidates $k$, the converter can be effectively utilized for cross-LLM inference [1].

# 4. Experiments

In this section, we present our experimental results. We begin with the experimental setup, followed by a comparison of watermark detection robustness and text quality against SOTA methods. We then discuss the differences between conventional KGW-type analytical watermarks and the proposed neural-based watermarking approach, followed by the ablation studies.

## 4.1. Experiment Settings

### 4.1.1. IMPLEMENTATION DETAILS

To train our end-to-end model, we choose *OPT-1.3B* as the online LLM to reduce training cost. We use samples from the *WikiText-103* dataset (Merity et al., 2017) as prompts for training and *C4* (Raffel et al., 2020) for evaluation. The context window is set to be $W = 10$, with the watermark strength $\delta = 1.25$ and $k = 20$ for the top-$k$ watermark logits empirically. We employ MGDA (Huo et al., 2024; Désidéri, 2012) to balance the detection and semantic objectives and use the Adam optimizer with a fixed learning rate of 1e-4 training with 35k steps. All experiments are conducted on one single NVIDIA RTX A6000 48G GPU. The evaluation is performed based on the MarkLLM[2] (Pan et al., 2024), an open-source tool for benchmarking LLM watermark methods. For further training and evaluation details, please refer to our code and Appendix C.

---

[1]Given $k = 20$, we calculate that the probability of more than half of the candidate tokens being misaligned is 0.7% using the binomial distribution.

[2]https://github.com/THU-BPM/MarkLLM

### 4.1.2. METRICS

Following the tradition of Pan et al. (2024); Zhang et al. (2024), we benchmark LLM watermarking methods across four key aspects: 1) Detection effectiveness of the human-written and clean watermarked text, measured by the F1 scores at optimal thresholds; 2) Detection robustness, evaluated by subjecting the watermarked text to 3 types of text modifications; 3) Text quality, assessed using the text perplexity as well as performance on downstream tasks including machine translation and code generation; and 4) Efficiency of the watermarking model, evaluated by measuring the time and GPU memory overhead for text generation and watermark detection presented in Appendix E.

### 4.1.3. COMPETITORS

We compare our method with the SOTA logits-based methods: KGW (Kirchenbauer et al., 2023), SIR (Liu et al., 2024b), and Unigram (Zhao et al., 2024), as well as distortion-free methods: Unbiased (Hu et al., 2024) and DiPmark (Wu et al., 2024). A comparison involve more competitors is presented in Appendix A.1.

## 4.2. Robustness Quality Benchmarking

Table 2 compares the robustness and quality of five methods across three LLMs, with Unigram and DiPmark serving as the strong baselines. The performance gap ($\Delta$) is presented at the bottom of the table. We use the first 30 tokens from the *C4* dataset (Raffel et al., 2020) as prompts, and generate 200 clean watermarked tokens as a watermark sample, with original human-written text serving as non-watermarked samples. Robustness is evaluated under edited conditions (SS, CP, PA in Table 2), while text quality is assessed using PPL with *Llama2-13B* as the oracle. Unbiased relies on prompts and LLM API access which constraints absent in our method and other baselines, making it less efficient. Notably, **our model is trained exclusively on *OPT-1.3B* and employs the converter (see Sec. 3.6) for cross-LLM infer-**

*Table 3.* Performance of our method on extra LLMs. CL: Clean sample; SS: Synonymous substitution; CP: Copy-paste attack; PA: Paragraphing; PPL: text perplexity; NWM: No watermark.

| LLM | Robustness (F1↑) | | | | Quality | |
| | CL | SS | CP | PA | PPL↓ | |
| | | Ours | | | Ours | NWM |
|---|---|---|---|---|---|---|
| *Mixtral-7B* | 0.990 | 0.970 | 0.987 | 0.916 | 10.219 | 8.711 |
| *Llama3-8B* | 0.998 | 0.990 | 0.990 | 0.934 | 7.256 | 5.964 |
| *Llama3.2-3B* | 0.997 | 0.995 | 0.993 | 0.947 | 7.599 | 6.301 |

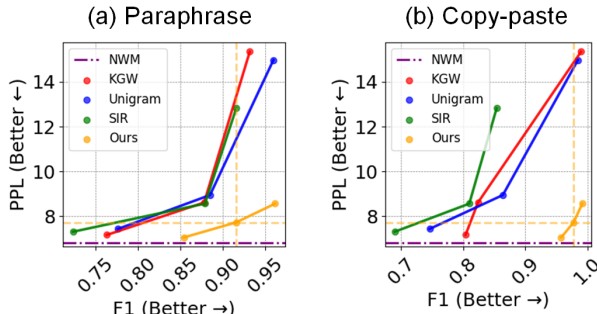

*Figure 5.* The trade-off between F1 score after text editing and PPL. (a) paraphrasing and (b) copy-paste attack.

*Table 4.* Downstream tasks performance. The best result is in **bold**, and the second-best is underlined. NWM: No watermark.

| Metric | NWM | KGW | Unigram | Unbiased | DiPmark | Ours |
|---|---|---|---|---|---|---|
| | Machine translation task with *NLLB-600M* | | | | | |
| BLEU↑ | 31.789 | 26.325 | 26.057 | 28.949 | 28.942 | **31.062** |
| | Code generation task with *Starcoder* | | | | | |
| pass@1↑ | 43.0 | 22.0 | 33.0 | **36.0** | **36.0** | 34.0 |

**ence.** Our method outperforms all baselines in robustness and quality across most scenarios, with minor degradations (1–3%) in a few cases, achieving an average F1 score of 0.975 across all scenarios. Our method achieves an average robustness improvement of 17.3% over DiPmark, with notable gains of 37–39% under paraphrasing, while maintaining comparable PPL scores. Compared to Unigram, our approach shows a 1.5% average robustness gain, peaking at 12% under copy-paste attacks for *Llama2-7B*, and reduces PPL by over 9% on average. These results highlight that our method effectively delivers watermark resilience while preserving LLM output quality, making it a practical solution for real-world applications.

To further validate the LLM-agnostic generalizability of our method, we conduct experiments on three additional LLMs, with the results shown in Table 3. Our approach consistently achieves a high F1 score (≥ 0.99) on clean watermarked samples across all LLMs, demonstrating its stability and reliability. Moreover, it maintains strong robustness against all editing types, yielding an average F1 score of 0.969. In terms of text quality, our method introduce only a moderate increase in PPL, approximately 1.2× that of non-watermarked baselines. These zero-shot results highlight the effectiveness of our approach without necessitating LLM-specific tuning, underscoring its broad applicability across diverse LLM architectures.

As shown in Table 4, we evaluate the impact of watermarking on downstream applications including machine translation (MT) and code generation (CG). Specifically, we assess translation performance using the WMT16 German-English benchmark with *NLLB-600M* and code generation using the HumanEval benchmark with *Starcoder*. For MT,

measured by BLEU score, our method achieves the highest score, 31.062, outperforming the second-best approach (Unbiased, 28.949) by 7.3% and significantly exceeding KGW and Unigram. In CG, evaluated by pass@1, our approach attains a competitive score of 34.0, closely trailing the distortion-free methods (DiPmark and Unbiased both 36.0) while surpassing Unigram and KGW. Notably, our approach exhibits exceptional robustness against distortion-free methods while achieving comparable quality, further demonstrating our superiority.

The strength parameter $\delta$ in logits-based methods is a critical factor in balancing robustness and quality. To systematically evaluate this trade-off, we conduct experiments assessing watermark resilience against paraphrasing and the copy-paste attack while measuring PPL. Watermarked text is generated under varying $\delta$ values[3]. The results, presented in Fig. 5, demonstrate the superior performance of our method in achieving an optimal balance between robustness and quality. At a PPL threshold of 8, where the non-watermarked baseline has a PPL of 6.8, our approach improves robustness by 18.16% under paraphrasing and 20.42% under copy-paste edits, surpassing the second-best method, Unigarm. In contrast, existing approaches such as KGW, Unigram, and SIR exhibit significant text quality degradation, with PPL values increasing by two to three times compared to the default $\delta$ while offering only marginal robustness improvements. These results highlight the effectiveness of our method in preserving text quality while achieving substantial robustness against text modifications.

Additional quantitative results are presented in Appendix A.

## 4.3. Comparsion to KGW-type Methods

We justify our proposed neural-based method by comparing to the KGW-type approaches, highlighting that our method offers guarantees analogous to provable p-values, which are

---

[3]Each method is tested at its default strength, as well as one lower and one higher setting: KGW and Unigram: {1, 2, 5}; SIR: {0.5, 1, 2}; Ours: {0.75, 1.25, 1.5}

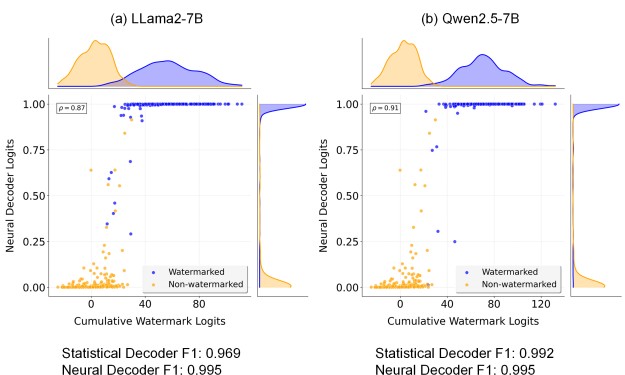

*Figure 6.* Correlation between the cumulative perturbed logits and the neural decoder logits of (a) *Llama2-7B* and (b) *Qwen2.5-7B* watermark samples.

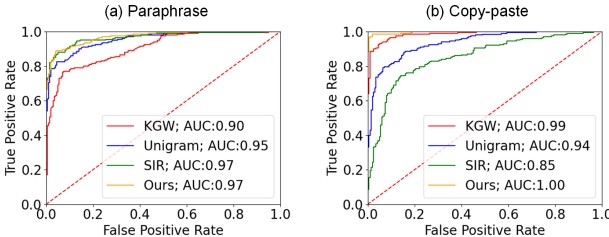

*Figure 7.* ROC curves for watermark detection under (a) paraphrasing and (b) copy-paste attack.

essential for text forensics.

**Red-Green List Retrieval.** Although our method does not explicitly rely on the p-value to determine the presence of the watermark, we allow for forensic analysis by retrieving the logits perturbation, which serves as the counterpart to the red-green list in KGW, for each token in a given sentence. Specifically, given an input sequence $[\mathbf{x}_1, \ldots, \mathbf{x}_n]$, tokenization is performed using the proposed converter. The retrieval is achieved by first obtaining the output distribution of $M$ at each step for $[\mathbf{x}_w, \ldots, \mathbf{x}_n]$ and subsequently reconstructing $\mathbf{S}_{\text{top } k}^{(t)}$ for each step $t$. The logits perturbation are then sequentially derived as $\mathbf{l}_{\text{top } k}^{(t)} = E(\mathbf{S}_{\text{top } k}^{(t)})$. Our watermarked samples colored with retrieval logits perturbation are visualized in Appendix B.1, this retrieval process enables the classification of tokens into green and red lists, thereby enhancing the interpretability of our watermark signal.

**Statistical Decoder vs. Neural Decoder.** Our method enables the retrieval of logits perturbation for a given sentence, allowing us to visualize the relationship between cumulative perturbed logits and neural decoder logits in Fig. 6. The analysis shows a strong positive correlation, where neural decoder logits increase as cumulative perturbed logits rise across the two LLMs. Additionally, the neural decoder outperforms the statistical decoder, likely due to the joint optimization process, which allows it to capture latent features

that enhance detection performance. The detection performance of our statistical and neural decoder is presented in the Appendix B.2.

**False Positive Thresholding.** Our model uses a neural decoder to generate a scalar confidence score for watermark detection. Similar to KGW-type methods, both employ a fixed threshold to detect watermarks, enabling adjustments to control false positive rates. Fig. 7 shows ROC curves under various text modifications, demonstrating the model's ability to manage false positive rates effectively. FPR across more LLMs is shown in Appendix A.5.

### 4.4. Ablation study

*Table 5.* Effect on different settings of our method. CL: Clean watermark sample; CP: Copy-paste attack; PA: Paragraphing; PPL: text perplexity; LD: log diversity.

| | *Llama2-7B* | | | | |
|---|---|---|---|---|---|
| Setting | Robustness | | | Quality | |
| | CL | PA | CP | PPL↓ | LD↑ |
| w/o $\mathcal{L}_{\text{sem}}$ | 0.995 | 0.960 | 0.982 | 9.561 | 8.165 |
| w/o $N$ | 0.992 | 0.867 | 0.952 | 7.820 | 7.788 |
| $\delta$=.75, $k$=20 | 0.960 | 0.854 | 0.958 | 7.057 | 7.694 |
| $\delta$=.75, $k$=40 | 0.957 | 0.832 | 0.957 | 7.278 | 8.253 |
| $\delta$=1., $k$=20 | 0.983 | 0.898 | 0.980 | 7.518 | 8.016 |
| $\delta$=1., $k$=40 | 0.980 | 0.884 | 0.978 | 7.981 | 7.995 |
| $\delta$=1.25, $k$=40 | 0.992 | 0.931 | 0.983 | 8.346 | 8.003 |
| $\delta$=1.5, $k$=20 | 0.995 | 0.962 | 0.992 | 8.566 | 7.771 |
| $\delta$=1.25, $k$=20 | 0.995 | 0.916 | 0.978 | 7.730 | 7.594 |

The ablation study in Table 5 evaluates the impact of key components and hyperparameters on model performance. Removing the semantic loss $\mathcal{L}_{\text{sem}}$ degrades text quality, increasing PPL by 23.6% while retaining strong robustness. Disabling the online editor $N$ significantly weakens resistance to text edits, with F1 scores dropping by 5.7% for paraphrasing and 2.7% for the copy-paste attack. Our model enables users to balance quality metrics flexibly by adjusting the top-$k$ logits tokens without retraining. Increasing $k$ from 20 to 40 enhances log diversity but results in a slight increase in perplexity. Adjusting $\delta$ reveals a clear trade-off between robustness and text quality. Lowering $\delta$ to 0.75 reduces F1 scores for paraphrasing to 0.854 but achieves the lowest PPL at 7.057. Conversely, increasing $\delta$ improves robustness, with F1 scores peaking at 0.962 for paraphrasing and 0.992 for the copy-paste attack at $\delta = 1.5$, though at the cost of higher PPL, a 21.3% increase compared to $\delta = 0.75$. The optimal balance is achieved at $\delta = 1.25$, where F1 scores for paraphrasing remain high at 0.916 while maintaining a moderate PPL of 7.730.

## 5. Conclusion

We introduce a novel logits-based end-to-end model, where encoder and decoder networks are jointly optimized to im-

prove detection robustness and text quality. We overcome the non-differentiability of the online text editor and semantic loss computation by using a novel online-prompting technique that leverages the on-the-fly LLM as a differentiable surrogate. Our method can be easily generalized to different LLMs.

## Acknowledgements

This work was supported in part by Macau Science and Technology Development Fund under SKLIOTSC-2021-2023, 0022/2022/A1, and 0119/2024/RIB2; in part by Research Committee at University of Macau under MYRG-GRG2023-00058-FST-UMDF; in part by the Guangdong Basic and Applied Basic Research Foundation under Grant 2024A1515012536.

## Impact Statement

Embedding resilient watermarks in AI-generated text transforms how we govern and trust automated content. Platforms can reliably verify authorship, deterring plagiarism and malicious reuse, while publishers gain confidence in sourcing and provenance. Content consumers benefit from clearer attribution, reducing the spread of misinformation. Regulators and standard-setting bodies acquire practical tools to enforce intellectual-property rights and audit generative systems. By elevating accountability and transparency across the AI ecosystem, the proposed watermark method approach paves the way for responsible innovation, bolsters public trust, and safeguards creative and journalistic integrity.

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

# Appendix

## Contents

# A. More Quantitative Results

## A.1. More Competitors

*Table 6.* Overall performance of the LLM watermark methods. CL: Clean watermark sample; SS: Synonymous substitution; CP: Copy-paste attack; PA: Paragraphing. Detection Performance is evaluated on *Llama2-7B*. PPL: Text perplexity with *Llama2-13B*; MT: Machine translation with *NLLB-600M*; CG: Code generation with *Starcoder*. The best is in **bold**, and the second-best is underlined.

| Method | F1↑ | | | | | PPL↓ | BLEU↑ | pass@1↑ |
| | CL | SS | CP | PA | Avg. | Qlt | MT | CG |
|---|---|---|---|---|---|---|---|---|
| KGW ($\delta$=2) | **1.000** | 0.980 | 0.824 | 0.878 | 0.920 | 8.658 | 26.325 | 22.0 |
| Unigram ($\delta$=2) | 0.997 | **0.985** | 0.864 | 0.885 | 0.933 | 9.275 | 26.057 | 33.0 |
| SWEET ($\delta$=2) | 0.997 | 0.959 | 0.952 | 0.817 | 0.931 | 8.522 | 27.916 | **36.0** |
| SIR ($\delta$=1) | 0.982 | 0.945 | 0.810 | 0.879 | 0.904 | 8.566 | 27.557 | 30.0 |
| TSW | **1.000** | 0.964 | 0.913 | 0.849 | 0.932 | 8.466 | 28.355 | 32.0 |
| Unbiased | 0.990 | 0.785 | 0.912 | 0.684 | 0.843 | **7.565** | 28.949 | **36.0** |
| DiPmark | 0.983 | 0.779 | 0.915 | 0.670 | 0.837 | 7.681 | 28.942 | **36.0** |
| Ours ($\delta$=1.25, $k$=20) | 0.995 | **0.985** | **0.978** | **0.916** | **0.969** | 7.730 | **31.062** | 34.0 |

We compare our method with extra three logits-based methods: SWEET (Lee et al., 2023), SIR (Liu et al., 2024b), and TSW (Huo et al., 2024). Table 6 presents the detection performance and text quality across different watermark methods. In terms of robustness, our method outperforms all competitors, achieving the highest average F1 score of 0.969, surpassing the second-best method, Unigram by 3.9%. Our method demonstrates exceptional resilience against the copy-paste and paraphrasing attacks, achieving 0.978 and 0.916, respectively, outperforming the strongest baseline (Unigram) in each cases by 13.2% and 3.5%. Furthermore, our method achieves the highest BLEU score of 31.062, exceeding the next-best approach (Unbiased, 28.949) by 7.3%, and attains a competitive pass@1 score of 34.0, outperforming Unigram by 3.0%. In contrast, methods (such as KGW and Unigram) exhibit strong robustness generally cost of increased PPL (8.658 and 9.275, respectively), whereas our method maintains a lower PPL pf 7.730, improving output quality by 16.7% compared to Unigram. These results highlight the effectiveness of our approach in achieving superior watermark robustness while preserving high text quality across diverse tasks.

## A.2. Effect on Sentence Length

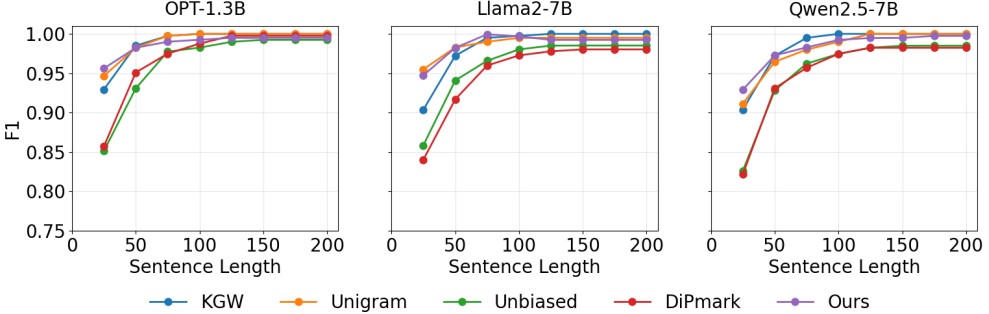

*Figure 8.* Detection performance (F1↑) on watermark sentences with various length.

Fig. 8 presents the F1 score of various watermarking methods across different sentence lengths for three LLMs: *OPT-1.3B*, *Llama2-7B*, and *Qwen2.5-7B*. The results demonstrate that across all methods, F1 scores generally improve as sentence length increases, indicating enhanced watermark detectability in longer text. Our method consistently achieves strong performance across all models and sentence lengths. Notably, for shorter sentences (length $\leq$ 50), our method attains higher F1 scores compared to other baselines, specially with an average improvement of 8% over DiPmark and Unbiased. For longer sentences (length $\geq$ 100), all methods converge toward near-optimal F1 scores ($\approx$ 1.00).

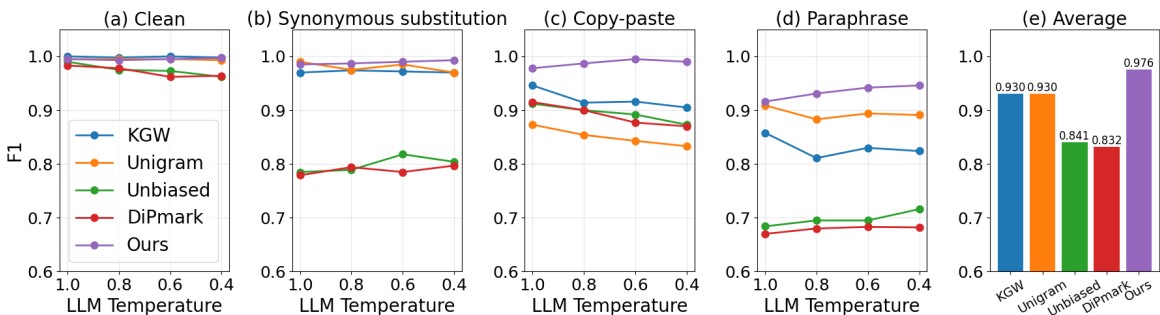

*Figure 9.* Detection performance (F1↑) on various LLM temperatures.

### A.3. Effect on LLM Temperature

Fig. 9 presents the F1 scores of different watermark methods across various LLM temperatures under clean and edited conditions, including synonymous substitution, copy-paste attack, and paraphrasing. The rightmost subfigure (e) reports the average F1 score across all conditions, highlighting overall robustness. Our method consistently outperforms all baselines, achieving an average F1 score of 0.976, which surpasses KGW and Unigram (both 0.930) by 4.9% and significantly outperforms Unbiased (0.841) and DiPmark (0.832) by 16.1% and 17.3%, respectively. Under clean conditions (a), all methods perform well, but our approach maintains a slight edge in stability across different temperatures. In synonymous substitution (b) and the copy-paste attack (c), our method demonstrates superior robustness, maintaining F1 scores near 1.0 while competing approaches experience noticeable declines. For paraphrasing (d), where detection is most challenging, our method exhibits a substantial advantage, outperforming the strongest baseline by over 7%. Lowering the temperature makes the LLM more deterministic, increasing the likelihood of sampling high-logit tokens. Since our model perturbs the top-logits, a lower temperature favors selecting tokens from our red/green lists over those in the grey list (see Section 3.2), thereby improving performance.

### A.4. Effect on LLM Sampling Strategy

*Table 7.* Detection performance (F1↑) of the LLM watermark methods with multinomial sampling and beam search. CL: Clean watermark sample; SS: Synonymous substitution; CP: Copy-paste attack; PA; Paragraphing. Detection Performance is evaluated on *Llama2-7B*. The best is in **bold**, and the second-best is underlined.

| Method | Multinomial sampling | | | | Beam search (num_beams=5) | | | |
|---|---|---|---|---|---|---|---|---|
| | CL | SS | PA | CP | CL | SS | PA | CP |
| KGW ($\delta$=2) | **1.000** | 0.980 | 0.878 | 0.824 | **1.000** | 0.998 | 0.940 | 0.975 |
| Unigram ($\delta$=2) | 0.997 | **0.985** | 0.885 | 0.864 | **1.000** | **1.000** | 0.954 | 0.939 |
| SIR ($\delta$=1) | 0.982 | 0.945 | 0.879 | 0.810 | 0.992 | 0.977 | 0.912 | 0.790 |
| Ours ($\delta$=1.25, $k$=20) | 0.997 | **0.985** | **0.916** | **0.978** | 0.997 | 0.995 | **0.962** | **0.995** |

Table 7 presents the performance of different watermarking methods under multinomial sampling and beam search (num_beams = 5). Our method achieves the best overall performance, demonstrating superior robustness in most scenarios. Under multinomial sampling, it outperforms all baselines in PA and CP, achieving F1 scores of 0.916 and 0.978, respectively, which are 3.5% and 13.2% higher than the second-best method (Unigram). For beam search, our method maintains strong robustness, achieving the highest F1 scores for PA (0.962) and CP (0.995), surpassing KGW by 2.3% and 2.1%, respectively. In contrast, while KGW and Unigram perform well under clean conditions (CL), with both achieving F1 scores of 1.000, they lag behind our method in edited scenarios. For instance, under CP with multinomial sampling, our method improves robustness by 13.5% compared to KGW. These results highlight the effectiveness of our approach in handling both clean and challenging text modifications, achieving a robust balance between accuracy and consistency.

*Table 8.* Best F1 FPR on clean watermark samples.

| LLM | Best F1 FPR↓ |
|---|---|
| *OPT-1.3B* | 0.00 |
| *Llama2-7B* | 0.03 |
| *Qwen2.5-7B* | 0.01 |
| *Mistral-7B* | 0.01 |
| *Llama3-8B* | 0.01 |
| *Llama3.2-3B* | 0.01 |
| Avg. | 0.01 |

### A.5. False Positive Rate on Clean Watermark Sentence

We present the best F1 FPR for various LLMs of our method in Table 8. Notably, *OPT-1.3B* achieves an FPR of 0.00, *Llama2-7B* is at 0.03, and all other models (*Qwen2.5-7B*, *Mistral-7B*, *Llama3-8B*, and *Llama3.2-3B*) consistently achieve an FPR of 0.01. With an average FPR of just 0.01, our method demonstrates exceptionally low false positive rates across different LLMs.

### A.6. Statistically Significant Improvement

*Table 9.* Paired t-test results comparing our model with competitors. Detection Performance is evaluated on *Llama2-7B*. PPL: Text perplexity with *Llama2-13B*. Mean Diff. represents the difference between our model and each competitor's mean performance. All p-values below 0.05 indicate statistically significant improvements.

| Method | PA(F1↑)-Trials | | | | | Mean ± Std | Mean Diff. | p-value |
|---|---|---|---|---|---|---|---|---|
| KGW ($\delta$=2) | 0.818 | 0.798 | 0.844 | 0.828 | 0.835 | 0.824 ± 0.018 | 0.087 | $2.12 \times 10^{-4}$ |
| Unigram ($\delta$=2) | 0.910 | 0.899 | 0.892 | 0.906 | 0.882 | 0.898 ± 0.011 | 0.014 | $3.80 \times 10^{-2}$ |
| Ours ($\delta$=1.25, $k$=20) | 0.918 | 0.908 | 0.909 | 0.906 | 0.917 | 0.912 ± 0.005 | - | - |
| Method | CP(F1↑)-Trials | | | | | Mean ± Std | Mean Diff. | p-value |
| KGW ($\delta$=2) | 0.843 | 0.832 | 0.845 | 0.847 | 0.839 | 0.841 ± 0.006 | 0.119 | $5.51 \times 10^{-6}$ |
| Unigram ($\delta$=2) | 0.864 | 0.900 | 0.859 | 0.861 | 0.866 | 0.870 ± 0.017 | 0.091 | $3.23 \times 10^{-4}$ |
| Ours ($\delta$=1.25, $k$=20) | 0.968 | 0.960 | 0.964 | 0.967 | 0.942 | 0.960 ± 0.011 | - | - |
| Method | PPL↓-Trials | | | | | Mean ± Std | Mean Diff. | p-value |
| KGW ($\delta$=2) | 8.656 | 8.678 | 8.800 | 8.730 | 8.409 | 8.655 ± 0.148 | -1.014 | $1.07 \times 10^{-4}$ |
| Unigram ($\delta$=2) | 8.871 | 8.832 | 8.918 | 9.155 | 9.092 | 8.973 ± 0.142 | -1.333 | $3.82 \times 10^{-6}$ |
| Ours ($\delta$=1.25, $k$=20) | 7.626 | 7.620 | 7.544 | 7.773 | 7.642 | 7.641 ± 0.083 | - | - |

To validate the statistical significance of our model's improvements, we conducted robustness and quality experiments using a paired t-test in Table 9. Specifically, we performed five trials to evaluate robustness against paraphrasing (PP) and copy-paste (CP) attacks, as well as the perplexity (PPL) of watermarked text. All trials were conducted using the same set of prompts, with multinomial sampling (same seed across methods in the same trial for consistency), resulting in diverse outcomes across the five trials. The watermark strength parameters ($\delta$) for KGW, Unigram, and our method were set to 2, 2, and 1.25, respectively, while the $k$-value for our method was set to 20. The results, shown in Table 9, indicate that our model achieves statistically significant improvements in both robustness and quality compared to the competitors. This is supported by p-values consistently below the 0.05 threshold.

# B. Watermark Analysis

## B.1. Watermark Sample

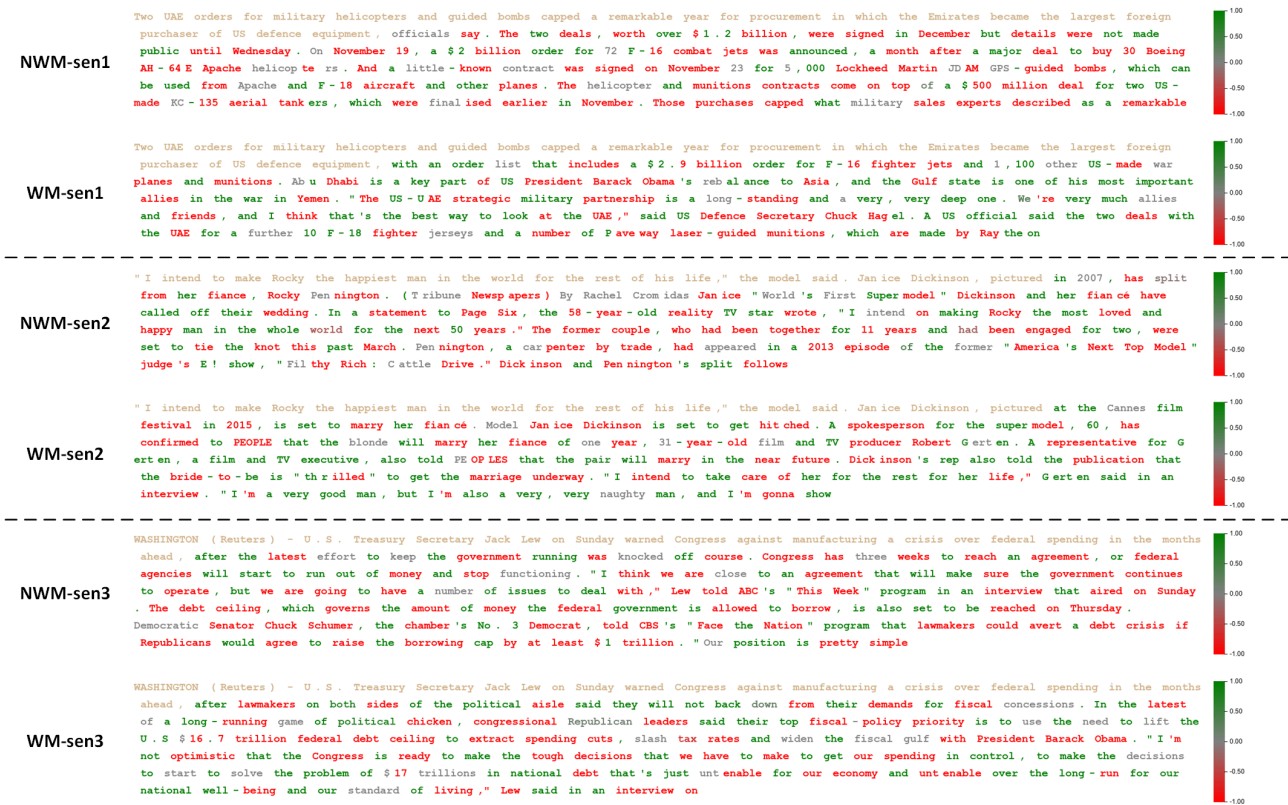

*Figure 10.* Visualizing three pairs of sentences comparing outputs from the original LLM (NWM) and the watermarked text (WM) of our method using the same prompt. Prompts are highlighted in gold, tokens are colored by the output values of the watermark encoder, and tokens outside the top-$k$ logits are shown in grey.

Fig. 10 presents three pairs of non-watermarked and watermarked samples generated from the same prompt of our method. The tokens are colored based on the output values retrieved from the watermark encoder. The watermarked samples maintain high quality while featuring a greater proportion of green-list tokens compared to red-list tokens. The ratios between NWM/WM texts are not similar at all, the NWM texts have red/green token counts of 69:53, 61:65, and 74:75 (ratios 1.30, 0.94, and 0.99). In contrast, the WM texts show counts of 42:83, 50:99, and 53:79 (ratios 0.51, 0.51, and 0.67). This is achieved by increasing the probabilities of green-list tokens during the generation process.

## B.2. Source of Robustness

We argue that robustness arises from two factors:

1. Watermark Decoder: By retrieving the red-green token lists with our watermark encoder (see Sec. 4.3), we can statistically compute token ratios that provide guarantees similar to provable p-values (analogous to KGW), which is critical for text forensics. As shown in Table 10, although our neural decoder—benefiting from end-to-end optimization with a noise layer—generally outperforms the statistical decoder, the latter remains competitive with strong baselines. At a fixed 1% FPR, the statistical decoder achieves a TPR of 0.96 in CP and a TPR of 0.65 in PA, compared to KGW's 0.81 (CP) and 0.50 (PA) as well as Unigram's 0.56 (CP) and 0.61 (PA). Since statistical decoder based solely on the red/green partition, and our neural decoder outperformed the statistical one, demonstrating that end-to-end training with a noise layer enhances robustness.

2. Red/Green Partition: We measure the KL divergence (KLD) between token distributions of watermark (WM) and non-watermark (NWM) sentences (see Fig. 11) to evaluate the context independence (CI) of our partition. A purely CI

Table 10. Detection performance of our neural decoder and statistical decoder.

| Method | 1%FPR TPR↑ | | | | Best F1↑ | | | |
|---|---|---|---|---|---|---|---|---|
| | CL | SS | CP | PA | CL | SS | CP | PA |
| KGW ($\delta = 2$) | 1.00 | 0.94 | 0.81 | 0.50 | 1.00 | 0.96 | 0.92 | 0.82 |
| Unigram ($\delta = 2$) | 1.00 | 0.98 | 0.56 | 0.61 | 1.00 | 0.98 | 0.85 | 0.92 |
| Ours (Neural Decoder, $\delta = 1.25, k = 20$) | 0.99 | 0.96 | 0.96 | 0.75 | 1.00 | 0.98 | 0.97 | 0.92 |
| Ours (Statistical Decoder, $\delta = 1.25, k = 20$) | 0.95 | 0.91 | 0.96 | 0.65 | 0.97 | 0.96 | 0.97 | 0.85 |

partition (e.g., Unigram) shows high KLD due to token biases, while our method achieves a KLD of 0.12—about half of Unigram's 0.21 and above KGW's 0.03—indicating a balance between context-dependent (CD) and CI schemes. Our ablation study on context size after paraphrasing is shown in Table 11. Our adaptive partition lets the encoder use strong CD features when available and fall back to a CI approach when necessary, which is crucial for our robustness.

Table 11. Detection Performance under different context size.

| Context Size | 1%FPR TPR↑ (PA) |
|---|---|
| 2 | 0.69 |
| 4 | 0.76 |
| 8 | 0.79 |
| 10 | 0.80 |

## B.3. Risk of Watermark Theft

**Acquisition of Different Watermarks.** It is NOT necessary to retrain the entire network from scratch for each user-specific watermark. The proposed end-to-end framework is flexible enough to inject key-driven randomness at multiple stages (input, model parameters, or output). In practice, one can fine-tune (FT) and apply key-conditioned post-processing without retraining. Different watermarks can be guaranteed by incorporating a key-driven bias logit $\mathbf{l}_B \in \{-1, 1\}^{|\mathcal{V}|}$ (of vocabulary size) into the final watermark logits via $\hat{\mathbf{l}} = \mathbf{l} + \delta \cdot (\mathbf{l}_W + \mathbf{l}_B)$, which $\mathbf{l}_B$ is only applied to the top-$k$ logits. Since $\mathbf{l}_B$ is derived from a unique key, statistical analysis demonstrates that each item in the clipped output remains unchanged with probability 0.5 and changes with probability 0.5 (assuming each $i$ item, $\mathbf{l}_W^{(i)} \sim \text{Uniform}(\{-1, 1\})$, $\mathbf{l}_B^{(i)} \sim \text{Uniform}(\{-1, 1\})$ and all items are independent). The number of mismatches between watermarks generated using different keys follows a binomial distribution $\text{Bin}(k, 0.5)$, implying that the probability of obtaining identical watermarks is negligibly small. E.g. $k = 20$, the probability of identical watermark is $0.5^{20} = 9.54 \times 10^{-7}$.

Table 12. 1%FPR TPR↓ of watermark theft on datasets Dolly CW and MMW BookReports.

| Method | Dolly CW | MMW BookReports |
|---|---|---|
| KGW | 0.45 | 0.59 |
| KGW (Diff. key) | 0.10 | 0.16 |
| Unigram | 0.22 | 0.04 |
| Unigram (Diff. key) | 0.11 | 0.15 |
| Ours (CD score) | 0.13 | 0.16 |
| Ours (CI score) | 0.55 | 0.56 |
| Ours (CI score, FT w/ $l_B$ 5k steps) | 0.19 | 0.12 |

We conduct a spoof attack (Jovanović et al., 2024) with 2,000 queries per method to evaluate resistance against watermark theft (lower TPR is better) in Table 12. Our method is more vulnerable with CI scoring (0.55 and 0.56 for the two datasets), but FT w/ reduces the TPR to 0.19 and 0.12, which is comparable to using different keys in KGW and Unigram.

Meanwhile, as shown in Table 13, our FT model shows performance comparable to the original checkpoint. Notably, the PPL improves from 7.73 to 7.28 but the TPR drops from 0.75 to 0.53 in the PA case.

*Table 13.* Overall performance of our finetune model with logtis bias.

| Method | 1%FPR TPR↑ | | | | PPL↓ | BLEU↑ | pass@1↑ |
|---|---|---|---|---|---|---|---|
| | CL | SS | CP | PA | Qlt | MT | CG |
| Ours | 0.99 | 0.97 | 0.98 | 0.75 | 7.73 | 31.06 | 34.00 |
| Ours (FT) | 1.00 | 0.98 | 0.99 | 0.53 | 7.28 | 30.94 | 32.00 |

## B.4. $\gamma$ of the Red-Green Split

*Table 14.* Statistics of $\gamma$ on different corpus.

| Hugging Face Dataset | Mean ± Std |
|---|---|
| xsum | 0.38 ± 0.11 |
| StackOverflow-QA-C-Language-40k | 0.43 ± 0.11 |
| ML-ArXiv-Papers | 0.40 ± 0.13 |
| mimic-cxr-dataset | 0.40 ± 0.12 |
| finance-alpaca | 0.41 ± 0.11 |

Different from KGW, the ratio of green tokens in the red-green partition $\gamma$ is not deterministic, but train by the watermark encoder, which could be vary with different context. We show the statistics of $\gamma$ in Table 14, and find that our method achieves an average $\gamma$ of 0.4 across datasets with diverse topics.

## B.5. Token Distribution Bias

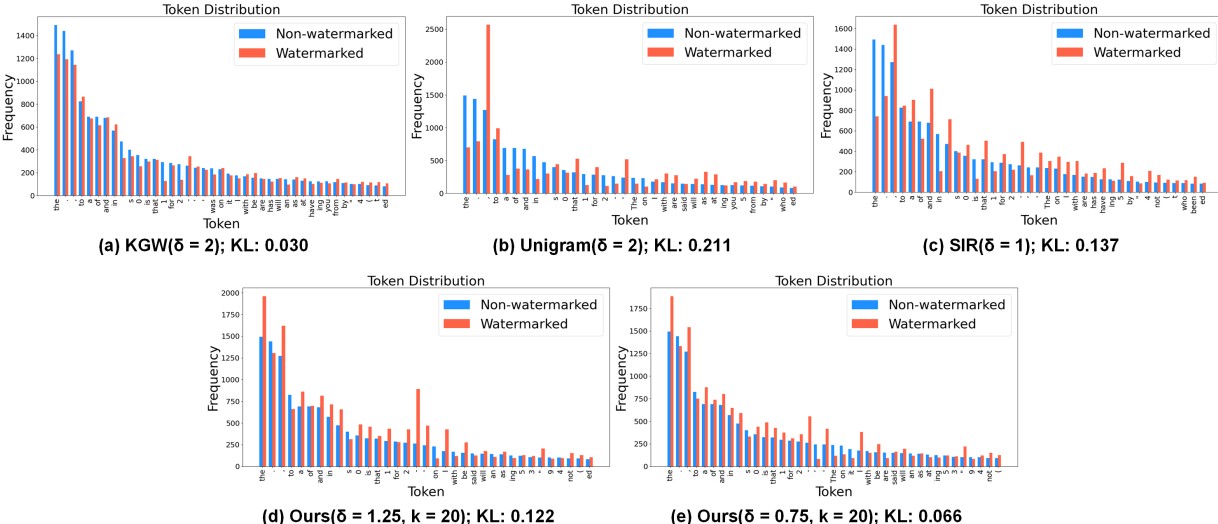

*Figure 11.* Token distribution and KD divergence (lower the better) between non-watermark/watermark sentences of our method in different settings as well as the SOTA competitors.

We further examine the effect of watermark strength $\delta$ on token distribution, as depicted in Fig. 11. With the default setting of $\delta = 1.25$, our model introduces less token bias compared to Unigram and SIR, while achieving superior robustness, as shown in Table 6. However, although our method outperforms KGW in robustness, it introduces a greater token bias. We argue that this highlights an inherent trade-off between robustness and token bias. Reducing $\delta$ to 0.75 reduces the KL divergence by 50% compared to the default setting, but comes at the expense of reduced robustness.

# C. Training and Evaluation Details

## C.1. Model Training Details

Table 15. Hyperparameters of the end-to-end model training.

| Item | Value |
|---|---|
| Learning rate | 1e-4 |
| Batch size | 8 |
| Training step | 35k |
| Encoder context size $w$ | 10 |
| On-the-fly LLM | OPT-1.3B |
| Top-$k$ candidate | 20 |
| Prompt tokens | 30 |
| Gumbel-softmax temperature $\tau_g$ | 0.1 |
| Probability of activating $N$ | 0.5 |
| Sharpness of $\tanh$, $\tau_t$ | 1000 |
| Maximum generated tokens | 100 |
| Watermark strength $\delta$ | 1 |
| Weight of $\mathcal{L}_{\text{dec}}$ | 10 |
| Weight of $\mathcal{L}_{\text{sem}}$ | 1 |

We trained our end-to-end model on a single NVIDIA RTX A6000 48GB GPU for 35k steps, completing the training in approximately 5 days with a GPU memory usage of 21.96GB. The hyperparameters used during training are detailed in Table 15. If GPU memory is limited, the batch size and maximum generated tokens can be reduced, or a smaller LLM, such as *OPT-125M*, can be used. While the training phase takes longer compared to existing training-based models like SIR, UPV, and TSW, this is due to the introduction of the entire LLM (with frozen parameters) in the training process. Despite the longer training time, we developed an efficient converter for cross-LLM inference, ensuring that the computational cost during inference remains low.

**Differentiability.** It is important to clarify that all prompts and generated text remain in the embedding domain throughout the training process. In our proposed online prompting, the prompt is first converted into the embedding domain and then concatenated with $\mathbf{X}_{\text{wm}}$ or $\mathbf{X}_{\text{nwm}}$. This ensures the entire process is differentiable, as we avoid the text-embedding transformation, which is the primary source of non-differentiability.

**Training Strategy.** To ensure stability and promote convergence in our end-to-end model, we adapt two training strategies: the Multiple-Gradient Descent Algorithm (Huo et al., 2024; Désidéri, 2012) and Curriculum Learning (Bengio et al., 2009).

1. **Multiple-Gradient Descent Algorithm.** The detection loss $\mathcal{L}_{\text{dec}}$ and the semantic loss $\mathcal{L}_{\text{sem}}$ are inherently conflicting: reducing one often increases the other (Huo et al., 2024). We apply MGDA to resolve this multi-objective optimization, which is proven to converge to a Pareto stationary solution (Désidéri, 2012).

2. **Curriculum Learning.** Training is divided into three progressive stages to move from easy to hard:

   - *Stage 1* ($< 10k$ steps): optimize only the detection objective.
   - *Stage 2* ($10k$–$20k$ steps): optimize both detection and semantic objectives jointly.
   - *Stage 3* ($> 20k$ steps): activate the noise layer for online paraphrasing together with the two objectives.

   This curriculum schedule enhances training stability and accelerates convergence.

## C.2. Training Stability

To assess training stability, we train our end-to-end model from scratch using three different seeds (5k steps each) and show the loss history and performance metrics in Fig. 12. We find that our models with different seeds converge in terms of detection loss and accuracy as the training process continues.

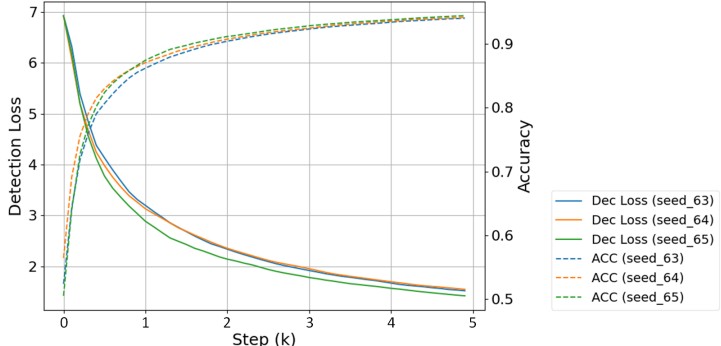

*Figure 12.* Training history of our model with three different seeds (63, 64, and 65).

*Table 16.* Editing configurations used in robustness evaluation.

| Edit | Description | Parameters |
|------|-------------|------------|
| Synonym substitute | Using BERT-large (Devlin, 2018) to select synonyms that fit the context. | Replace ratio: 0.5 |
| Paraphrasing | Paraphrasing the text while maintaining the original meaning with the deep paraphrase model Dipper(Krishna et al., 2023). | Lex diversity: 60 Order diversity: 0 |
| Copy-paste | Concatenating the watermarked text after the human-written text creates a mix with only parts of the watermarked text. | Watermark segment ratio: 25% |

### C.3. Editing Configuration

We compare the robustness of watermark methods across 3 types of editing. The setting of all edits strictly followed the open-source studies, MARKLLM[4] and SIR[5]. For transparency, we provide a detailed description in Table 16, parameter setting for each text editing.

### C.4. Examples of online text modification

Although our training pipeline primarily uses only paraphrasing for text editing, we demonstrate the potential of online text modification when integrating the LLM into the proposed end-to-end training framework. By leveraging online prompting techniques, more advanced editing can be achieved. Table 17 provides examples of three more common editing types: random token drop, synonym substitution, and copy-paste attack, along with their corresponding prompts, original texts, and edited outputs.

---

[4]https://github.com/THU-BPM/MarkLLM
[5]https://github.com/THU-BPM/Robust_Watermark

Table 17. Examples of online text modification.

| Editing Type | Prompt | Original text | Revised text |
|---|---|---|---|
| Random Drop Tokens | Please randomly drop some words from the below given text: | In a significant leap forward for artificial intelligence (AI) technology, researchers at NeuralNet Innovations have developed an advanced algorithm that promises to enhance decision-making capabilities in AI systems, potentially revolutionizing industries from healthcare to autonomous vehicles. | In leap forward for AI technology, researchers at NeuralNet Innovations developed advanced algorithm that promises enhance decision-making capabilities in systems, potentially revolutionizing industries from healthcare to autonomous vehicles. |
| Synonym Substitute | Please randomly select some words from the below given text and substitute them with their synonyms: | Mental health is a critical component of overall well-being and has a significant impact on individuals' ability to function effectively in their daily lives, including at work. Employers play a vital role in promoting mental health by creating supportive environments that encourage open communication, reduce stigma, and provide resources for employees. | Mental health is an essential element of general well-being and has a substantial influence on individuals' capacity to operate efficiently in their everyday lives, including at work. Employers play a crucial role in advancing mental health by establishing supportive atmospheres that encourage candid communication, diminish prejudice, and offer assets for staff. |
| Copy-paste | Please randomly copy the below given text and paste to the provided human-written text to form the new text: | Climate change is one of the most pressing environmental issues of our time, posing significant threats to biodiversity worldwide. | Global warming is a critical issue that affects various aspects of our planet. Climate change is one of the most pressing environmental issues of our time, posing significant threats to biodiversity worldwide. It leads to melting ice caps, rising sea levels, and more frequent natural disasters. These changes have profound impacts on both human societies and wildlife. ... |
| Paragraphing | Rewrite the following paragraph: | We introduce a novel logits-based end-to-end model, where encoder and decoder networks are jointly optimized to improve detection robustness and text quality. Constructing such a system is challenging due to the non-differentiability of key modules, such as the online text editor and semantic loss computation. | We propose a novel logits-based end-to-end model that jointly optimizes encoder and decoder networks to enhance detection robustness and text quality. Developing this system poses significant challenges, particularly due to the non-differentiability of critical components, including the online text editor and semantic loss computation. |

## D. Comparison with Post-Generation Watermarking Methods

Post-generation watermarking methods, such as AWT (Abdelnabi & Fritz, 2021) and REMARK-LLM (Zhang et al., 2024), embed watermarks after the text has been fully generated. These approaches rely on a language model to rephrase the generated text, embedding a watermark signal while preserving the semantic meaning of the original sentences. However, post-generation methods have notable limitations. They do not fully leverage the capabilities of the original LLM and are more susceptible to out-of-distribution (OOD) issues. For instance, models trained on datasets like HC3, a natural language question-answering corpus, often struggle with OOD inputs, such as code, leading to reduced performance on tasks like code generation (e.g., lower code passing scores). In contrast, logits-based methods, including ours, embed watermarks during the generation process by sampling tokens directly from a perturbed distribution. This approach minimally constrains the LLM, allowing it to retain its natural understanding of language while maintaining broad compatibility across diverse tasks.

## E. Watermark Efficiency

Table 18. Time and memory consumption for generation and detection with a 200-token sample of our method.

| LLM | Setting | Generation | | Detection | |
|---|---|---|---|---|---|
| | | Time↓ (s) | Memory↓ (GB) | Time↓ (s) | Memory↓ (GB) |
| *OPT-1.3B* | w/o watermark | 2.557 | 5.900 | 0.003 | 0.008 |
| *OPT-1.3B* | w/ watermark | 2.769 | 5.900 | 0.003 | 0.008 |
| *Llama2-7B* | w/o watermark | 5.204 | 15.793 | 0.005 | 0.008 |
| *Llama2-7B* | w/ watermark | 8.362 | 15.793 | 0.005 | 0.008 |

In Table 18, we assess the computational time overhead and GPU memory usage of our method. For *OPT-1.3B*, the lightweight encoder design results in only an 8.3% increase in generation time for watermarked text, with no change in maximum GPU memory usage since the encoder is invoked after each token generation. For *Llama2-7B*, our method increases the generation time by 60.6%, mainly due to the embedding transformation from *Llama2-7B* to *OPT-1.3B*, as the tokenizer cannot be accelerated by the GPU and is called at each step. The time overhead for watermark embedding can be mitigated through parallel tokenization, reducing the time complexity by up to $1/k$. In terms of watermark detection, our decoder operates efficiently, requiring only negligible time and memory consumption.

Table 19. Detection time (s) on a single NVIDIA A6000 GPU for a 200-token sample.

| Method | Detection Time↓ (s) |
|---|---|
| KGW | 0.3 |
| EXP-Edit | 80 |
| Unbiased | 3.4 |
| Ours | 0.005 |

Moreover, our watermark detection is extremely efficient. As shown in Table 19, with *Llama2-7B* using a single NVIDIA A6000 GPU, our method requires only 0.005 seconds per watermarked sample (200 tokens), compared to 0.3 seconds for KGW, 3.4 seconds for Unbiased, and 80 seconds for EXP-Edit. Overall, our method is 16,000 times faster than EXP-Edit and 680 times faster than Unbiased, making it highly suitable for scalable watermarking systems.

