# OpenReview forum: "An End-to-End Model for Logits-Based Large Language Models Watermarking"
_ICML.cc/2025/Conference — ICML 2025 poster_

### Official Review · Reviewer_eHFF · 2025-03-12

**Overall Recommendation:** 2

**Summary:**

This paper proposes an end-to-end model for logits-based LLM watermarking. The model consists of a logit perturbation generation network and a watermark detection network, and these two networks are trained in an end-to-end pipeline. To improve the robustness of watermark, a LLM is used to paraphrase watermarked text, and the detection network are trained to identify the modified watermarked text and non-watermarked text. To improve the quality of watermarked text, a LLM is used to extract the semantic embeddings of watermarked text and non-watermarked, and the cosine distance between the two embeddings is minimized. Experiments compare the detection performance of the watermark text of this method and multiple key-based methods after various tampering, as well as the performance of the watermark text in downstream tasks. Experimental results show the proposed method achieves better robustness and text quality.

**Claims And Evidence:**

This paper argues that the proposed end-to-end neural network-based watermarking method is robust and has little impact on text quality. The training method and objective function design in this chapter include these two goals, and the experimental results show that the method is effective in both aspects. However, this paper lacks theoretical justification for the source of watermark robustness, relying primarily on empirical results.

**Essential References Not Discussed:**

One key contribution of this paper is maintaining text quality of watermarked LLM, but the state-of-the-art method on this topic, namely SynthID published in Nature does not been cited and compared. However, it is understandable given that SynthID was published in October 2024, only about 3 months before the conference deadline.

**Experimental Designs Or Analyses:**

The paper contains several experiments to demonstrate the proposed method achieves the two objectives, maintaining text quality and robustness against text modifications.
1)	For text quality, the paper evaluates the proposed with multiple key-based watermarking methods on the perplexity of watermarked text and the metrics (BLEU, pass@1) of downstream tasks including translation and code generation.
2)	For robustness, the paper evaluates the proposed method with multiple key-based watermarking methods on clean watermarked text and corrupted watermarked text under synonymous substitution, copy-paste attack and paragraphing to demonstrate its ability against tampering.

A critical concern about the experiments is that this part lacks experiments and analyses on resisting watermark stealing attacks. Additionally, in the watermarking scenario, measuring the True Positive Rate (TPR) at fixed False Positive Rate (FPR) thresholds would be more informative than using F1 scores alone, as this better reflects real-world requirements where a certain level of false alarms can be tolerated while maximizing detection capability.

**Methods And Evaluation Criteria:**

This paper experimentally evaluates the robustness and quality of generated text of the proposed method compared with the baseline methods. In the key-based watermarking method, the context window size is directly related to the robustness. This paper lacks ablation experiments on the context window size, making the source of the robustness of the watermarking method unclear. In the key-based watermarking method, a watermark with strong robustness is more easily stolen as its watermark pattern space is relatively small, but this problem can be solved by using multiple keys, because different keys can easily create independent watermark patterns. This paper does not explore the risk of the watermark pattern being stolen by the proposed watermarking method and how to mitigate this risk.

**Other Comments Or Suggestions:**

1.	Improve transparency of methodology by extracting and analyzing the red-green split and logits perturbations produced by the model to more clearly illustrate the source of the robustness of this method.
2.	Discuss how to mitigate the risk of watermark stealing.
3.	Discuss how to obtain diverse watermarks efficiently without requiring complete retraining of the model.

**Other Strengths And Weaknesses:**

Strengths:
1.	This paper develops an end-to-end neural network-based watermarking method to achieve a better balance between watermark robustness and maintaining text quality.
2.	This paper proposes two techniques to improve watermark performance and practice model training. One is to let the classifier distinguish the watermarked text paraphrased by LLM, and the other is to use LLM to generate the semantic embeddings of watermarked text and non-watermarked text.

Weaknesses:
1.	The source of the robustness of the watermarking method in this paper is unclear. It may stem from the ability of detector, or the watermark pattern, or both. For watermark pattern, this method does not constrain the diversity of red-green partitions or the proportion of green tokens. An extreme example is that the neural network may produce a globally invariant red-green partition similar to UniGram.
2.	The author did not discuss the risk of the proposed watermarking being stolen. Generally, high robustness will bring a higher risk of watermark stealing. The key-based method can alleviate this problem by changing the key, but the flexibility of the end-to-end architecture in this paper is weaker, resulting in this risk that cannot be ignored.
3.	High computational cost and low flexibility. The key-based watermarking method can obtain independent watermarking patterns at zero cost by rotating the key. The watermarking based on the end-to-end model requires high computational cost, and it is difficult to guarantee the acquisition of independent and different watermarking modes even after retraining, which limits its practical application scope.

**Questions For Authors:**

1.	For key-based watermark, different pseudorandom number generator with different keys can provide different watermark for different users. For this end-to-end watermark, there are two questions: I) To create different watermarks for different users, should the networks be train from the beginning to end? II) How to guarantee different watermarks trained from the same neural network framework are distinguishable?
2.	How are the differences in the red-green partition and logits perturbation of the watermark patterns generated by the proposed method for different contexts? For example, what is the ratio of green tokens in the red-green partition for different contexts, and how is the difference in logits perturbation?
3.	In Figure 10 of Appendix E, why are the ratios of green tokens and red tokens similar for the non-watermarked text and the watermarked text?
4.	Figure 9 of Appendix C shows that as the LLM temperature decreases, the F1 of the watermark detection of some watermarking methods decreases. It stands to reason that as the temperature decreases, the uncertainty/diversity of the text generated by LLM will increase, and the increase in text entropy will improve the watermark effect. The experimental results here are inconsistent with this conclusion. What do the authors think is the reason for this phenomenon? Is this because the watermark method applies the logits/probability distribution before the temperature processing, rather than after the temperature processing?

**Relation To Broader Scientific Literature:**

This article proposes a watermarking method based on neural networks and red-green lists, which achieves better text generation quality and robustness. The red-green list-based watermarking method comes from the literature [1] (Kirchenbauer et al). It uses a pseudo-random number generator with a key to use context tokens as seeds to divide the vocabulary into two sets. By adding a small perturbation (delta) to the logits, the probability of some tokens (called green tokens, with a ratio of gamma) is increased and the probability of another token (called red tokens) is reduced. In the detection stage, by observing the proportion of green tokens in the text and the z-score, it is determined whether the text has a watermark. The key and pseudo-random number generator ensure the unpredictability of the watermark. Replacing different keys can generate independent red and green lists, which increases the security of the watermark.
Reference [2] (Zhao et al.) proposed a robust watermarking method by using a fixed red-green list during text generation, and the authors suggest changing keys to maintain watermark security.
Reference [3] (Jovanović et al.) proposed a method for stealing watermarks (green lists) based on statistical analysis. The conclusion of the reference is that the watermark pattern generated by a single key is easy to be stolen, and using multiple keys to generate watermark text in rotation can alleviate this problem.
Reference [4] (Huo et al.) proposed using a neural network to adjust the proportion of green tokens and the perturbation intensity (delta) of logits based on the use of a pseudo-random number generator to divide the vocabulary. This approach achieves a better balance between watermark detectability and semantic preservation. The contribution of reference [4] is most similar to this paper. However, reference [4], based on the characteristics of the pseudo-random number generator, faces a smaller risk of watermark stealing compared to the current paper's end-to-end neural approach.
Overall, using neural networks to improve robustness and text generation quality is feasible, as demonstrated in this paper. However, the authors appear to have overlooked the increased security risks associated with their approach, particularly the vulnerability to watermark stealing attacks that may be more pronounced in end-to-end neural models compared to key-based methods with flexible pattern switching capabilities.

[1] Kirchenbauer, John, et al. "A watermark for large language models." International Conference on Machine Learning. PMLR, 2023.
[2] Zhao, Xuandong, et al. "Provable Robust Watermarking for AI-Generated Text." The Twelfth International Conference on Learning Representations.
[3] Jovanović, Nikola, Robin Staab, and Martin Vechev. "Watermark Stealing in Large Language Models." International Conference on Machine Learning. PMLR, 2024.
[4] Huo, Mingjia, et al. "Token-specific watermarking with enhanced detectability and semantic coherence for large language models." Proceedings of the 41st International Conference on Machine Learning. 2024.

**Theoretical Claims:**

refer to the question and weakness part

---

> ### Author Rebuttal · Authors · 2025-03-30
>
> Dear Reviewer eHFF,
>
> We sincerely appreciate the time and effort you have dedicated. Below, we summarize the key responses to your concerns.
>
> # Source of Robustness
> We argue that robustness arises from two factors:
>
> 1.**Watermark Decoder**: We compare our neural decoder (ND) with a statistical decoder (SD) based solely on the red/green partition (see response to Reviewer sPDj), and our ND outperformed SD, demonstrating that end-to-end training with a noise layer enhances robustness.
>
> 2.**Red/Green Partition**: We measure the KL divergence (KLD) between token distributions of watermark (WM) and non-watermark (NWM) sentences (see [here](https://postimg.cc/RWN7WHMx)) to evaluate the context independence (CI) of our partition. A purely CI partition (e.g., Unigram) shows high KLD due to token biases, while our method achieves a KLD of 0.12—about half of Unigram’s 0.21 and above KGW’s 0.03—indicating a balance between context-dependent (CD) and CI schemes. Our ablation study on context size after paraphrasing is shown below:
>
> |Context Size|1%FPR TPR↑ (PA)|
> |-|-|
> |2|0.69|
> |4|0.76|
> |8|0.79|
> |10|0.80|
>
> Our adaptive partition lets the encoder use strong CD features when available and fall back to a CI approach when necessary, which is crucial for our robustness.
>
> ---
>
> # Acquisition of Different Watermarks
> It is **NOT** necessary to retrain the entire network from scratch for each user-specific watermark. The proposed end-to-end framework is flexible enough to inject key-driven randomness at multiple stages (input, model parameters, or output). In practice, one can fine-tune (FT) and apply key-conditioned post-processing without retraining.
>
> Different watermarks can be guaranteed by incorporating a key-driven bias logit $l_B \in \lbrace-1,1\rbrace^n$ (of vocabulary size) into the final watermark logits via $ \hat{l}_W = \delta \times \text{clip}(l_W + l_B) $. Since $l_B$ is derived from a unique key, statistical analysis demonstrates that each item in the clipped output remains unchanged with probability 0.5 and changes with probability 0.5 (assuming the $i$ item, $\text{clip}(l_W^{(i)} + l_B^{(i)}) \sim \text{Uniform}(-1, 1)$, $l_B^{(i)} \sim \text{Uniform}(\lbrace-1, 1\rbrace)$, trainable $l_W^{(i)} \in [-1,1]$, and all items are independent). The number of mismatches between watermarks generated using different keys follows a binomial distribution $\text{Bin}(n, 0.5)$, implying that the probability of obtaining identical watermarks is negligibly small. E.g. $n = 40$, the probability of identical is $0.5^{40} \approx 9.1\times 10^{-13}$. The explanation (with the spoof attack below), which highlights the randomness introduced by the key, will be included in the final version.
>
> ---
>
> # Risk of Watermark Stolen
> |1%FPR TPR↓|Dolly CW|MMW BookReports|
> |-|-|-|
> |KGW|0.45| 0.59|
> |KGW (Diff. key)|0.10|0.16|
> |Unigram|0.22|0.04|
> |Unigram (Diff. key)|0.11|0.15|
> |Ours (CD score)|0.13|0.16|
> |Ours (CI score)|0.55|0.56|
> |Ours (CI score, FT w/ $l_B$ 5k steps)|0.19|0.12|
>
> We conduct a spoof attack with 2,000 queries per method to evaluate resistance against watermark stolen (lower TPR is better). Our method is more vulnerable with CI scoring (0.55 and 0.56 for the two datasets), but FT w/ $l_B$ reduces the TPR to 0.19 and 0.12, which is comparable to using different keys in KGW and Unigram.
>
> | |1%FPR TPR↑ (CL / SS / CP / PA)|Qlt  (PPL↓ / BLEU↑ / pass@1↑)|
> |-|-|-|
> |Ours|0.99 / 0.97 / 0.98 / 0.75|7.73 / 31.06 / 34.00|
> |Ours (FT)|1.00 / 0.98 / 0.99 / 0.53|7.28 / 30.94 / 32.00|
>
> Meanwhile, our FT model shows performance comparable to the original checkpoint. Notably, the PPL improves from 7.73 to 7.28 but the TPR drops from 0.75 to 0.53 in the PA case.
>
> ---
>
> # $\gamma$ of Red-Green Split
> |Hugging Face Dataset|$\gamma$|
> |-|-|
> |xsum|0.38±0.11|
> |StackOverflow-QA-C-Language-40k|0.43±0.11|
> |ML-ArXiv-Papers|0.40±0.13|
> |mimic-cxr-dataset|0.40±0.12|
> |finance-alpaca|0.41±0.11|
>
> Our method achieves an average $\gamma$ of 0.4 across datasets with diverse topics.
>
> ---
>
> # Compare with SynthID
> | |1%FPR TPR↑ (CL / SS / CP / PA)|Qlt (PPL↓ / BLEU↑ / pass@1↑) |
> |-|-|-|
> |SynthID|0.99 / 0.61 / 0.29 / 0.05|5.71 / 26.13 / 37.00|
> |Ours|0.99 / 0.97 / 0.98 / 0.75|7.73 / 31.06 / 34.00|
>
> Our method consistently outperforms SynthID in editing cases, notably scoring 0.75 versus SynthID's 0.05 in PA, while maintaining competitive quality.
>
> ---
>
> # Ratio of Red/Green tokens in Fig 10
> The ratios between NWM/WM texts are not similar at all, the NWM texts have red/green token counts of 69:53, 61:65, and 74:75 (ratios 1.30, 0.94, and 0.99). In contrast, the WM texts show counts of 42:83, 50:99, and 53:79 (ratios 0.51, 0.51, and 0.67).
>
> ---
>
> # Fig 9 Explanation
> Lowering the temperature makes the LLM more deterministic, increasing the likelihood of sampling high-logit tokens. Because our model perturbs the top-$k$ logits, a lower temperature favors selecting tokens from our red/green lists over those in the grey list (see Sec. 3.2), thereby improving performance.

---

### Official Review · Reviewer_bhtZ · 2025-03-13

**Overall Recommendation:** 4

**Summary:**

The paper introduces a logits-based end-to-end model for watermarking LLM generated text. As the existing method can not achieve an optimal balance between text quality and robustness, the authors propose a novel approach that jointly optimizes encoder and decoder to improve both text quality and robustness. Experiments show the proposed method achieves superior robustness while maintaining comparable text quality.

Strength:
1. End-to-End Optimization: Jointly optimizes encoder and decoder for better alignment and efficiency.
2. Superior Robustness: Achieves higher resistance to tampering compared to existing methods.
3. Balanced Performance: Maintains text quality while significantly improving robustness.

Weakness:
1. Lack some case studies. If the author can add some specific case studies, that would be much better.
2. Author can include more LLM, besides OPT and llama2.

**Claims And Evidence:**

The claims about improved robustness, maintained text quality, and cross-LLM generalizability are well-supported by the empirical results.

**Essential References Not Discussed:**

NO

**Experimental Designs Or Analyses:**

Maybe the training stability or variance in model performance across different random initializations should be discussed.

**Methods And Evaluation Criteria:**

The methods and evaluation criteria are well-designed for LLM watermarking, with appropriate consideration for both technical performance (robustness) and practical usability (quality preservation across diverse tasks).

**Other Comments Or Suggestions:**

NO

**Other Strengths And Weaknesses:**

NO

**Questions For Authors:**

NO

**Relation To Broader Scientific Literature:**

NO

**Theoretical Claims:**

Its contributions are primarily empirical and validated through extensive experimental evaluation, not theoretical analysis.

---

> ### Author Rebuttal · Authors · 2025-03-30
>
> Dear Reviewer bhtZ,
>
> We sincerely appreciate the time and effort you have dedicated to our manuscript. Below, we summarize the key responses to your concerns.
>
> # Case Study
>    We have included three pairs of non-watermarked and watermarked samples generated from the same prompt in Appendix E (see [here](https://postimg.cc/hQ09L2Wv)), where tokens are color-coded based on the watermark encoder's outputs.
>
> ---
>
> # Additional LLMs
>
> |LLM|Best F1↑ (CL / SS / CP / PA)|Qlt (NWM-PPL↓ / Ours-PPL↓)|
> |-|-|-|
> |Qwen2.5-7B|1.00 / 0.99 / 0.99 / 0.95|8.92 / 10.00|
> |Mixtral-7B|1.00 / 0.97 / 0.99 / 0.92|8.71 / 10.22|
> |Llama3-8B|1.00 / 0.99 / 0.99 / 0.93|5.96 / 7.26|
> |Llama3.2-3B|1.00 / 1.00 / 0.99 / 0.95|6.30 / 7.60|
>
> In addition to OPT-1.3B and Llama2-7B, we have evaluated our method on four extra LLMs—Qwen2.5-7B, Mixtral-7B, Llama3-8B, and Llama3.2-3B—for robustness and quality (in Sec. 4.2). Our approach consistently achieves a high F1 score (≥ 0.99) on
> clean watermarked samples across all LLMs, demonstrating its stability and reliability. Moreover, it maintains strong robustness against all edited cases, yielding an average F1 score ≥ 0.92. In terms of text quality, our method introduces only a moderate increase in PPL, approximately 1.2× that of non-watermarked baselines.
>
> |Method|Qlt (NLLB-600M-BLEU↑ / Starcoder-pass@1↑)|
> |-|-|
> |NWM|31.79 / 43.0|
> |KGW|26.33 / 22.0|
> |Unigram|26.06 / 33.0|
> |Unbiased|28.95 / 36.0|
> |DiPmark|28.94 / 36.0|
> |Ours|31.06 / 34.0|
>
> We have also assessed translation quality using NLLB-600M and code generation with Starcoder (in Sec. 4.2). For the machine translation task, measured by BLEU score, our method achieves the highest score, 31.062, outperforming the second-best approach (Unbiased, 28.949) by 7.3%. In the code generation task, evaluated by pass@1, our approach attains a competitive score of 34.0, closely trailing the distortion-free methods (DiPmark and Unbiased both 36.0) while surpassing Unigram and KGW. Notably, our approach exhibits exceptional robustness against distortion-free methods while achieving comparable quality, further demonstrating our superiority.
>
> ---
>
> # Training Stability
>    To assess training stability, we will train our end-to-end model from scratch using three different seeds (5k steps each) and will share the loss history and performance metrics once training is complete (the entire training process will take about 60 hours).

---

### Official Review · Reviewer_sPDj · 2025-03-14

**Overall Recommendation:** 3

**Summary:**

The authors propose a method to enhance the robustness of logit-based watermarking techniques while preserving text quality. The main idea is to use a model to generate the "biases" for the logits.

**Claims And Evidence:**

The claims are generally supported by enough evidence.

**Essential References Not Discussed:**

N/A

**Experimental Designs Or Analyses:**

The experimental setup is generally sound.

**Methods And Evaluation Criteria:**

The methods and evaluation criteria are suitable for the problem.

**Other Comments Or Suggestions:**

In section 4.1.3, the citations are missing a space.

Line 803 is missing space, too (Table11 --> Table 11)

**Other Strengths And Weaknesses:**

The method moderately improves the robustness-text quality tradeoff compared to the baselines. However, I wouldn’t say the results are groundbreaking.

A few weaknesses I noticed are: (1) The method loses the theoretical guarantees of prior works, such as KGW, regarding false positive rates; (2) The method requires additional compute compared to prior work: the watermark biases predictor needs to be trained, and there is a significant overhead during inference, especially on Llama2-7B.

**Questions For Authors:**

What is the false positive rate of the method when it is not under any attack?

**Relation To Broader Scientific Literature:**

The paper builds on state-of-the-art watermarking methods, such as KGW. Its distinction is that it does not split the vocabulary randomly but instead learns the biases for the tokens, which they show improves the robustness - text quality tradeoff.

**Theoretical Claims:**

N/A

---

> ### Author Rebuttal · Authors · 2025-03-31
>
> Dear Reviewer sPDj,
>
> We sincerely appreciate the time and effort you have dedicated to our manuscript. Below, we summarize the key responses to your concerns.
>
> # Theoretical Guarantees
>
> ||1%FPR TPR↑ (CL / SS / CP / PA)|Best F1↑ (CL / SS / CP / PA)|
> |-------------------|---------------------------------|---------------------------------|
> | KGW|1.00 / 0.94 / 0.81 / 0.50| 1.00 / 0.96 / 0.92 / 0.82|
> | Unigram|1.00 / 0.98 / 0.56 / 0.61| 1.00 / 0.98 / 0.85 / 0.92|
> | Ours (Neural Decoder)| 0.99 / 0.96 / 0.96 / 0.75| 1.00 / 0.98 / 0.97 / 0.92|
> | Ours (Statistical Decoder)|0.95 / 0.91 / 0.96 / 0.65| 0.97 / 0.96 / 0.97 / 0.85|
>
> By retrieving the red-green token lists with our watermark encoder (see Sec. 4.3), we can statistically compute token ratios that provide guarantees similar to provable p-values (analogous to KGW), which is critical for text forensics. Although our neural decoder—benefiting from end-to-end optimization with a noise layer—generally outperforms the statistical decoder, the latter remains competitive with strong baselines. For example, at a fixed 1% FPR, the statistical decoder achieves a TPR of 0.96 in CP and a TPR of 0.65 in PA, compared to KGW’s 0.81 (CP) and 0.50 (PA) as well as Unigram’s 0.56 (CP) and 0.61 (PA).
>
> ---
>
> # Additional Computational Overhead
>
> Although our model requires extra resources during end-to-end training, we developed a converter that enables efficient, cross-LLM inference without retraining. The additional time shown in Table 11 is due to the converter applying two tokenizers to each of the \(k\) input sequences; however, parallel LLM inference (batching prompts) can significantly reduce this overhead.
>
> |Method|Detection Time↓ (seconds)|
> |-|-|
> |KGW|0.3|
> |EXP-Edit|80|
> |Unbiased|3.4|
> |Ours|0.005|
>
> Moreover, our watermark detection is extremely efficient. On Llama2-7B using a single NVIDIA A6000 GPU, our method requires only 0.005 seconds per watermarked sample, compared to 0.3 seconds for KGW, 3.4 seconds for Unbiased, and 80 seconds for EXP-Edit. Overall, our method is 16,000 times faster than EXP-Edit and 680 times faster than Unbiased, making it highly suitable for scalable watermarking systems.
>
> ---
>
> # False Positive Rate Under No Attack
>
> |LLM|Best F1 FPR↓|
> |-|-|
> | OPT-1.3B|0.00|
> | Llama2-7B|0.03|
> | Qwen2.5-7B|0.01|
> | Mistral-7B|0.01|
> | Llama3-8B|0.01|
> | Llama3.2-3B|0.01|
> | Average|0.01|
>
> We present the best F1 FPR for various LLMs. Notably, OPT-1.3B achieves an FPR of 0.00, Llama2-7B is at 0.03, and all other models (Qwen2.5-7B, Mistral-7B, Llama3-8B, and Llama3.2-3B) consistently achieve an FPR of 0.01. With an average FPR of just 0.01, our method demonstrates exceptionally low false positive rates across different LLMs.
>
> ---
>
> # Typo
>
> The missing spaces of the citations in Sec. 4.1.3 and line 803 will be fixed in the final version.

---

### Decision · Program_Chairs · 2025-05-01

**Decision:**

Accept (poster)

**Comment:**

The paper proposes an end-to-end model to watermark LLM generated text. The model consists of a logit perturbation generation network and a watermark detection network, and these two networks are trained in an end-to-end pipeline. To improve the robustness of watermark, a LLM is used to paraphrase watermarked text, and the detection network are trained to identify the modified watermarked text and non-watermarked text. To improve the quality of watermarked text, a LLM is used to extract the semantic embeddings of watermarked text and non-watermarked, and the cosine distance between the two embeddings is minimized. Experiments compare the detection performance of the watermark text of this method and multiple key-based methods after various tampering, as well as the performance of the watermark text in downstream tasks. Experimental results show the proposed method achieves better robustness and text quality.

Strengths:
- The paper jointly optimizes encoder and decoder to improve watermarking quality and robustness.
- The paper proposes two techniques to improve watermark performance and practice model training.

Weakness:
- The paper lacks a theoretical analysis of generation quality, detection rate, and robustness. It is unclear whether the original KGW guarantees still hold.

Reviewers' additional concerns regarding fine-tuning cost, source of robustness, experiments of watermark stealing attacks are addressed by the authors during rebuttal.